# CORAL: Disentangling Latent Representations in Long-Tailed Diffusion

**Esther Rodriguez**
Arizona State University
`earodr32@asu.edu`

**Monica Welfert**
Arizona State University
`mwelfert@asu.edu`

**Samuel McDowell**
Arizona State University
`scmcdowe@asu.edu`

**Nathan Stromberg**
Arizona State University
`nstrombe@asu.edu`

**Julian Antolin Camarena**
Arizona State University
`jantolin@asu.edu`

**Lalitha Sankar**
Arizona State University
`lsankar@asu.edu`

## Abstract

Diffusion models have achieved impressive performance in generating high-quality and diverse synthetic data. However, their success typically assumes a class-balanced training distribution. In real-world settings, multi-class data often follow a long-tailed distribution, where standard diffusion models struggleproducing low-diversity and lower-quality samples for tail classes. While this degradation is well-documented, its underlying cause remains poorly understood. In this work, we investigate the behavior of diffusion models trained on long-tailed datasets and identify a key issue: the latent representations (from the bottleneck layer of the U-Net) for tail class subspaces exhibit significant overlap with those of head classes, leading to feature borrowing and poor generation quality. Importantly, we show that this is not merely due to limited data per class, but that the relative class imbalance significantly contributes to this phenomenon. To address this, we propose **CO**ntrastive **R**egularization for **A**ligning **L**atents (CORAL), a contrastive latent alignment framework that leverages supervised contrastive losses to encourage well-separated latent class representations. Experiments demonstrate that CORAL significantly improves both the diversity and visual quality of samples generated for tail classes relative to state-of-the-art methods. The implementation code is available at `https://github.com/SankarLab/coral-lt-diffusion`.

## 1 Introduction

Diffusion models (DMs) [1, 2] have achieved impressive performance in generating high-quality and diverse samples across a range of domains. However, their success typically relies on class-balanced training data. In practice, many real-world datasets exhibit *long-tailed* class distributions, where a small number of head classes contain the majority of samples, while many tail classes are significantly underrepresented [3]. Under such imbalance, DMs often fail to generate faithful and diverse outputs for tail classes, instead exhibiting feature borrowing, where samples from rare classes display a mix of tail and head features [4].

Recent work has sought to improve generative models under long-tailed class distributions by addressing sampling imbalance and promoting class-aware generation. Class-Balancing Diffusion Models (CBDMs) [5] introduce a regularizer that encourages balanced sampling across classes by penalizing deviations from a target distribution. In particular, the approach enhances tail generation based on the model prediction on the head class. This increased reliance on the model prediction and conditional priors introduces bias and can potentially reduce robustness (*e.g.*, lead to class entanglement) during training. To address these limitations, Zhang et al. [6] propose a Bayesian

39th Conference on Neural Information Processing Systems (NeurIPS 2025).

framework with weighted denoising score-matching and a gating mechanism to selectively transfer information from head to tail classes. Other works incorporate contrastive learning [7], a key technique in metric learning [8], to improve class separability. For example, Yan et al. [4] propose a probabilistic contrastive approach that reduces overlap among class-conditional distributions to enhance tail-class generation. While these approaches have improved performance in imbalanced settings, they *primarily operate in the ambient image space or introduce latent representations external to the denoising process*. In contrast, relatively little attention has been given to structuring class representations within the latent space of the denoising network itself, highlighting a key gap in current methods.

The core generative process relies on a neural architecture that processes data through a lower-dimensional latent space. Current models predominantly use the U-Net architecture [2, 9], which incorporates an encoder-bottleneck-decoder structure consisting of convolutional neural networks to downsample, followed by a multilayer perceptron, and then upsampling by further convolutional neural networks back into image space. There are skip connections from the encoder to the decoder to preserve data and feature information and mitigate vanishing gradients. It has been shown that the U-Net's bottleneck output carries semantic meaning [10]. One of our key observations is that, under long-tailed distributions, tail-class samples tend to occupy regions in this latent space that overlap heavily with head classes. This overlap — which we refer to as *representation entanglement* — undermines model's the ability to preserve class-specific features, leading to poor generative performance for tail classes. We base this observation on extensive visualizations of long-tailed datasets for diffusion using various distance-preserving mappings. Figure 1 illustrates this effect for t-SNE [11] and shows how tail-class representations are absorbed into dominant clusters.

To address this, we propose **CO**ntrastive **R**egularization for **A**ligning **L**atents (CORAL), a contrastive latent alignment method that operates directly on the latent representations within the denoising network. Inspired by metric learning and its applications to learning representations [12, 13, 14], CORAL augments the encoder of the denoising U-Net with a *projection head* applied to the bottleneck output. The resulting projected embeddings are trained using a supervised contrastive loss, which is then combined with the standard diffusion objective. This encourages the model to pull together representations of samples from the same class while pushing apart those from different classes, thereby promoting class-wise separation in the latent space. In contrast to prior work that applies contrastive losses in the ambient or auxiliary latent spaces, CORAL regularizes the internal feature space of the diffusion model itselfprecisely where representation entanglement arises.

Our contributions are summarized as follows:

- **Empirical analysis of long-tailed diffusion behavior:** We provide evidence that diffusion models trained on long-tailed data are prone to representation entanglement in the latent space of the denoising U-Net, particularly at the bottleneck layer, which contributes to low-quality tail-class generation.

- **Identification of representation entanglement as a root cause:** We show that the generation failure for tail classes stems from entanglement in the models latent feature representations induced by severe class imbalance, revealing a previously unexplored failure mode.

- **Proposal of CORAL:** We introduce **CO**ntrastive **R**egularization for **A**ligning **L**atents (CORAL), a contrastive latent alignment method that encourages separation between class-wise latent representations by augmenting the diffusion model with a supervised contrastive loss applied to projected bottleneck features.

- **Improved tail-class generation:** Through extensive experiments on several long-tailed datasets (CIFAR10-LT, CIFAR100-LT [15], CelebA5 [16], ImageNet-LT[17]), we demonstrate that CORAL significantly improves both the diversity and visual fidelity of tail-class samples, outperforming prior approaches. Moreover, we provide qualitative and quantitative evidence that CORAL promotes class-wise separation in the latent space of the denoising network, directly addressing the class entanglement that impairs tail-class generation.

## 1.1 Related Work

**Diffusion Models for Imbalanced Data**    Standard methods for class-conditioned diffusion sampling include classifier guidance (CG) [19], which requires a separately trained classifier, and classifier-free guidance (CFG) [18], which jointly trains conditional and unconditional denoisers. While widely

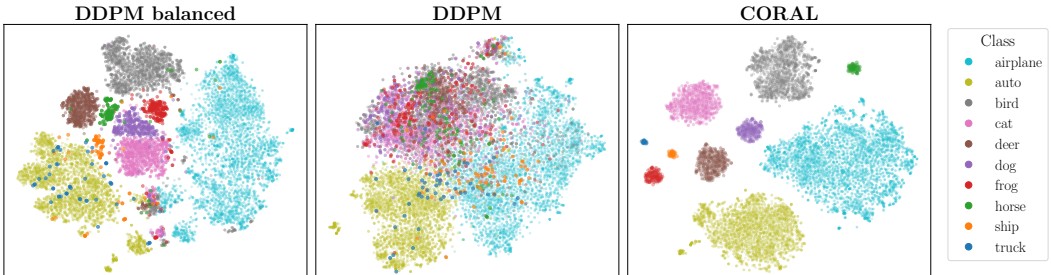

Figure 1: **t-SNE visualizations of U-Net bottleneck features**. The dataset visualized is CIFAR10-LT where the tail-to-head ratio is $0.01$, *i.e.,* the head class (airplane) is 100 times more represented than the tail class (truck), with an exponential decay in-between. Real CIFAR10-LT samples are passed through models trained under different settings. Shown are **(left)** DDPM [18] trained on the original balanced CIFAR-10 dataset, **(middle)** DDPM trained on CIFAR10-LT with an imbalance ratio of $0.01$, and **(right)** CORAL trained under the same imbalanced setting. In the balanced case, class representations are moderately separated, though some overlap remains. Under imbalance, DDPM exhibits substantial overlap between head and tail classes, an effect we refer to as *representation entanglement*, which degrades generation quality for tail classes. CORAL mitigates this effect by promoting class-wise separation in the latent space.

used, both CG and CFG struggle to generate diverse, high-quality samples for underrepresented tail classes [4].

Several recent approaches have been proposed to improve tail-class performance in diffusion models, most of which operate in the ambient (image) space. For example, Class-Balancing Diffusion Models (CBDMs) [5] introduce a regularizer that penalizes deviations from a balanced class distribution, guiding the model to allocate more capacity to underrepresented classes during training. Time-dependent importance weighting [20] adjusts the loss based on sampling time to mitigate bias, while oriented calibration [6] uses Bayesian gating mechanisms to transfer knowledge from head to tail classes (H2T) during unconditional generation and from tail to head (T2H) during conditional generation. DiffROP [4] applies a contrastive regularization based on KL divergence to reduce class-conditional overlap at the output level.

In contrast to these ambient-space approaches, CORAL operates directly in the latent space of the diffusion model. Specifically, CORAL introduces a supervised contrastive loss on projected bottleneck features from the denoising U-Net, encouraging class-wise separation through metric learning. This latent-space regularization provides a more direct and structured means of disentangling class representations.

Relatedly, Han et al. [21] propose LDMLR, which generates synthetic latent features for long-tailed datasets using a DDIM trained on encoder representations from a fixed model. While effective for long-tailed recognition, LDMLR operates as a post hoc feature augmentation method and does not modify the generative process. For the same objective of long-tailed recognition, Shao et al. [22] use a chosen classifier's feature space to guide the diffusion model for the tail classes and filter out out-of-distribution samples during generation. In contrast, CORAL directly regularizes the latent space during training, promoting class separation within the diffusion model without relying on a separate inference model.

**Contrastive Learning in Latent-Variable Generative Models**   Contrastive learning (CL) is a widely adopted technique for structuring embedding spaces in supervised, self-supervised, and metric learning settings [7, 8]. Recent work has extended CL to generative models: DiffROP [4] applies a probabilistic contrastive loss in the ambient space to reduce overlap between class-conditional output distributions in diffusion models; CONFORM [23] introduces contrastive regularization over attention maps to improve semantic alignment in text-to-image generation. TVAE [13] and Tri-VAE [14] both incorporate a triplet loss into a variational autoencoder (VAE) framework: TVAE for general representation learning and Tri-VAE for anomaly detection. While TVAE uses a standard VAE architecture, Tri-VAE employs a U-Net with a projection head at the bottleneck, similar in spirit to CORAL. However, neither method involves diffusion and both modify the decoder path.

## 2 Preliminaries and Problem Setup

### 2.1 Diffusion Models

Generative DMs were first introduced in [1], which formulated data generation as a Markovian denoising process grounded in non-equilibrium thermodynamics. The approach was later popularized by Ho et al. [2], who introduced a simplified objective and fixed variance schedule, significantly improving sample quality and training stability.

DMs generate data by gradually adding noise to a sample in a forward (noising) process and then learning to denoise in a reverse (denoising) process. The forward process gradually corrupts a data sample $\mathbf{x}_0 \sim q(\mathbf{x}_0)$ over $T$ discrete time steps by adding Gaussian noise:

$$q(\mathbf{x}_t|\mathbf{x}_{t-1}) = \mathcal{N}(\mathbf{x}_t; \sqrt{1-\beta_t}\mathbf{x}_{t-1}, \beta_t\mathbf{I}) \quad \text{and} \quad q(\mathbf{x}_{1:T}|\mathbf{x}_0) = \prod_{t=1}^{T} q(\mathbf{x}_t|\mathbf{x}_{t-1}), \quad 1 \le t \le T,$$

(1)

where $\{\beta_t \in (0,1)\}_{t=1}^{T}$ is a predefined variance schedule. As $t$ increases, the distribution of $\mathbf{x}_t$ transitions from close to $q(\mathbf{x}_0)$ to approximately standard Gaussian. One can express the marginal distribution at any timestep $t$ as $q(\mathbf{x}_t|\mathbf{x}_0) = \mathcal{N}(\mathbf{x}_t; \sqrt{\bar{\alpha}_t}\mathbf{x}_0, (1-\bar{\alpha}_t)\mathbf{I})$, where $\bar{\alpha}_t = \prod_{s=1}^{t}(1-\beta_s)$.

The reverse process is modeled by a neural network that approximates the conditional distribution $q(\mathbf{x}_{t-1}|\mathbf{x}_t)$ using a learnable Gaussian distribution given by

$$p_\theta(\mathbf{x}_{t-1}|\mathbf{x}_t) = \mathcal{N}(\mathbf{x}_{t-1}; \boldsymbol{\mu}_\theta(\mathbf{x}_t, t), \boldsymbol{\Sigma}_\theta(\mathbf{x}_t, t)) \quad \text{and} \quad p_\theta(\mathbf{x}_{0:T}) = p(\mathbf{x}_T)\prod_{t=1}^{T} p_\theta(\mathbf{x}_{t-1}|\mathbf{x}_t), \quad (2)$$

where the terminal distribution is typically set to $p(\mathbf{x}_T) = \mathcal{N}(\mathbf{0}, \mathbf{I})$ and the variance is often fixed as $\boldsymbol{\Sigma}_\theta(\mathbf{x}_t, t) = \sigma_t^2\mathbf{I}$. Rather than predicting $\boldsymbol{\mu}_\theta$ directly, it is common to reparameterize the mean in terms of the added noise $\boldsymbol{\epsilon}_\theta(\mathbf{x}_t, t)$ and train the network to predict this noise.

This leads to the simplified training objective introduced by [2], in which the model $\boldsymbol{\epsilon}_\theta(\mathbf{x}_t, t)$ is trained to predict the Gaussian noise $\boldsymbol{\epsilon} \sim \mathcal{N}(\mathbf{0}, \mathbf{I})$ that was used to perturb the clean input $\mathbf{x}_0$ into a noised version $\mathbf{x}_t$, for a randomly sampled timestep $t \sim \mathcal{U}(\{1, \ldots, T\})$:

$$\mathcal{L}_{\text{diff}} = \mathbb{E}_{t, \mathbf{x}_0, \boldsymbol{\epsilon}} \left[ ||\boldsymbol{\epsilon} - \boldsymbol{\epsilon}_\theta(\mathbf{x}_t, t)||_2^2 \right]. \tag{3}$$

In practice, the noise predictor $\boldsymbol{\epsilon}_\theta$ is implemented using an image-to-image U-Net parameterized by $\theta$, and the expectation operator is replaced by the empirical sample average for every $t$.

**Classifier-Free Guidance** When label information is available, classifier-free guidance (CFG) [18] has become a widely adopted technique for improving conditional diffusion models. Instead of training a separate classifier to guide generation, CFG modifies the denoising model $\boldsymbol{\epsilon}_\theta$ in (3) to support both conditional and unconditional generation. During training, the model $\boldsymbol{\epsilon}_\theta(\mathbf{x}_t, t, \mathbf{y})$ is optimized using class labels $\mathbf{y}$; for a chosen fraction $p_{\text{uncond}}$ of samples, the training process ignores labels to learn the unconditional model with $\mathbf{y} = \varnothing$.

Finally, at sampling time, conditional guidance is applied by combining the conditional and unconditional predictions to recover $\mathbf{x}_0$ from $\mathbf{x}_t$:

$$\boldsymbol{\epsilon}_\theta^{\text{CFG}}(\mathbf{x}_t, t, \mathbf{y}) = (1+\omega)\boldsymbol{\epsilon}_\theta(\mathbf{x}_t, t, \mathbf{y}) - \omega\boldsymbol{\epsilon}_\theta(\mathbf{x}_t, t), \tag{4}$$

where $\omega > 0$ is a guidance weight controlling the strength of conditioning.

### 2.2 Metric Learning and Contrastive Approaches

Metric learning aims to map inputs into an embedding space where semantically similar examples are close together and dissimilar ones are far apart. Instead of designing distance functions manually, modern methods use neural networks to learn transformations that make standard distances (*e.g.*, Euclidean or cosine) meaningful for the task. This learned embedding captures complex similarity structures aligned with supervision.

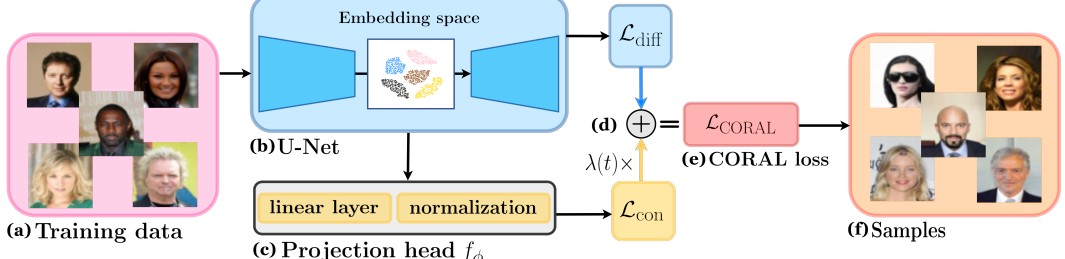

Figure 2: **CORAL architecture and workflow on CelebA-5.** **(a)** The five-class CelebA-5 training data is input to the U-Net architecture. **(b)** Denoising U-Net. The white inset shows an actual t-SNE visualization of the U-Net latent representations due to CORAL. **(c)** CORAL's addition to the standard DDPM architecture: a projection head MLP consisting of a single dense layer followed by normalization. **(d)** The output from the U-Net and the projection head are used to compute the corresponding diffusion and contrastive losses. **(e)** The contrastive loss is scaled by a time-dependent weighting function, $\lambda(t)$, and added to the standard diffusion loss to obtain the CORAL loss. **(f)** Samples are obtained from a trained CORAL model.

### 2.2.1 Contrastive Loss Functions

Contrastive loss functions are a fundamental tool in metric learning, designed to shape embedding spaces so that semantically similar samples are close together, while dissimilar samples are pushed apart. Early formulations, such as the triplet loss [24], enforce a margin between anchor-positive and anchor-negative pairs using triplets of labeled samples where each triplet contains an anchor, a positive (same class as anchor), and a negative (different class from anchor). While effective, triplet loss can suffer from slow convergence and inefficient sampling. More recent advancements such as the supervised contrastive loss (SupCon) [25] generalize this idea by leveraging all positives and negatives in a mini-batch, offering greater stability and improved sample efficiency during training.

**Supervised Contrastive Loss**  SupCon [25] generalizes triplet loss by comparing each anchor to multiple positives and negatives within a batch, improving both convergence stability and overall performance. Let $\mathbf{z} \in \mathbb{R}^d$ denote the $\ell_2$-normalized embedding of a sample. The loss is defined as:

$$\mathcal{L}_{\text{SupCon}} = -\sum_{i \in I} \frac{1}{|P(i)|} \sum_{p \in P(i)} \log \frac{\exp(\mathbf{z}_i \cdot \mathbf{z}_p / \tau_{\text{SC}})}{\sum_{s \in S(i)} \exp(\mathbf{z}_i \cdot \mathbf{z}_s / \tau_{\text{SC}})} \tag{5}$$

where $I$ is the set of all indices in the batch, $S(i) = I \setminus \{i\}$ denotes the set of all sample indices in the batch excluding the anchor $i$, $P(i) \subseteq S(i)$ is the set of indices corresponding to positive samples that share the same class as the anchor, and $\tau_{\text{SC}}$ is a temperature parameter that controls the concentration (sharpness) of the similarity distribution. Lower values of $\tau_{\text{SC}}$ (*e.g.*, $\tau_{\text{SC}} \approx 0.1$) sharpen the distribution, placing greater emphasis on harder positive and negative pairs and increasing the gradient magnitude ($|\nabla \mathcal{L}_{\text{SC}}| \propto 1/\tau_{\text{SC}}$).

## 3  Our Method

In this section, we present our method, **CO**ntrastive **R**egularization for **A**ligning **L**atents (CORAL), designed to enhance class separation in diffusion models trained on long-tailed datasets. The core insight behind CORAL is that the latent space of the denoising U-Net, specifically its bottleneck layer, plays a central role in shaping generative behavior. In long-tailed settings, we observe that latent representations of tail-class samples often overlap with those of head classes, resulting in *representation entanglement* and degraded generation quality (see Figure 1 for a visualization of the CIFAR10-LT dataset). Our comparisons between models trained on balanced and imbalanced data indicate that this overlap arises from head classes dominating parameter updates, resulting in less structured latent representations for tail classes.

CORAL introduces two targeted modifications to standard diffusion training: a lightweight projection head applied to the U-Net bottleneck and a supervised contrastive loss term. These additions allow

CORAL to regularize the latent space directly, promoting intraclass clustering and interclass separation during training. The contrastive signal complements the diffusion objective, helping maintain semantic distinctions across classes, especially for underrepresented ones. Figure 2 illustrates the CORAL framework using CelebA-5 [16], a 5-class LT subset of CelebA [26], as an example dataset.

**Architectural Modification**   While the forward diffusion process operates in the high-dimensional ambient space, the U-Net processes information through a compressed latent space, with the bottleneck layer playing a central role in the model's representational capacity. Prior work has shown that this bottleneck encodes semantically meaningful features [10], making it a natural point for intervention. CORAL leverages this insight by adding a small projection head $f_\phi$, *e.g.* a fully-connected linear layer followed by a normalization layer, to the bottleneck output.

CORAL builds on established principles of contrastive representation learning, where projection heads have been shown to capture task-related information more effectively than direct feature space constraints [12, 27]. The projection head serves two critical functions: (1) it decouples the contrastive objective from the main diffusion features, allowing the model to learn class-discriminative embeddings in an auxiliary space while the bottleneck continues to serve the generative objective [12], and (2) it preserves intraclass diversity by preventing the contrastive loss from directly collapsing intraclass variations in the bottleneck representations [27]. This complementary structure allows both the diffusion and contrastive objectives to improve simultaneously.

During training, we apply a supervised contrastive loss on the projected embeddings to encourage class-wise separation while the main bottleneck features continue to serve the diffusion objective. Once trained, the U-Net bottleneck has learned to produce well-separated class-specific features, and the projection head is no longer needed. This design ensures that CORAL adds zero computational overhead during sampling and remains fully compatible with standard diffusion sampling procedures. The learned structure in the bottleneck representations persists and guides generation without requiring the auxiliary projection network.

Table 4 in Appendix B demonstrates that CORAL's latent space intervention consistently outperforms ambient space approaches in the style of DiffROP [4]. Ambient-space approaches enforce separation on already generated outputs, which can reduce intraclass diversity by imposing constraints post-hoc. In contrast, CORAL learns inherently separated representations at the compressed bottleneck layer where class overlap occurs during the generative process itself.

**Training Objective**   The overall training objective for CORAL augments the standard diffusion loss with a contrastive alignment term applied to the projected latent representations to obtain

$$\mathcal{L}_{\text{CORAL}} = \mathcal{L}_{\text{diff}} + \lambda(t) \cdot \mathcal{L}_{\text{con}}, \tag{6}$$

where $\mathcal{L}_{\text{diff}}$ is the standard diffusion training loss, such as the noise prediction objective defined in (3), $\mathcal{L}_{\text{con}}$ is a contrastive loss applied to the projected bottleneck features, and $\lambda(t)$ is a time-dependent weighting function. While we use $\mathcal{L}_{\text{con}} = \mathcal{L}_{\text{SupCon}}$ in our experiments, the framework is general and supports any contrastive loss. The weighting function $\lambda(t)$ is defined as:

$$\lambda(t) = w \cdot \exp\left(\frac{1 - t/T}{\tau_r}\right), \quad t \in \{0, 1, \ldots, T\} \tag{7}$$

where $w$ is the base contrastive weight, $T$ is the total number of diffusion steps, and $\tau_r$ is the temperature parameter that controls the decay rate. Although in general $\tau_r > 0$, our results with the SupCon loss suggest that a range between $[0.5, 1.0]$ works best.

This dynamic weighting scheme places greater emphasis on the contrastive objective during the earlier (less noisy, $t \approx 0$) denoising steps, where a meaningful semantic structure is more recoverable, and gradually reduces its influence at later steps ($t \approx T$), where noise dominates the input. This encourages more discriminative latent representations during the most informative stages of training.

**Training Procedure**   To train our proposed CORAL method, we modify the standard diffusion training procedure to incorporate both contrastive latent regularization and classifier-free guidance. Algorithm 1 summarizes the full training pipeline. For each mini-batch, we first sample diffusion timesteps and generate noisy inputs via the standard DDPM forward process. In line with CFG training protocol [18], we randomly drop class labels with a fixed probability to enable joint training

of conditional and unconditional denoising. The noisy inputs and (possibly masked) labels are passed through the U-Net to compute the standard diffusion loss. Simultaneously, we extract bottleneck features from the U-Net encoder, project them via the projection head $f_\theta$, and compute a supervised contrastive loss using the original (unmasked) class labels. We then compute the total loss $\mathcal{L}_{\text{CORAL}}$ given in (6) Model parameters are updated using backpropagation on $\mathcal{L}_{\text{CORAL}}$.

---

**Algorithm 1** CORAL Training Procedure

---

**Input:** Dataset $\mathcal{D}$, model $\epsilon_\theta$, projection head $f_\phi$, total diffusion steps $T$, guidance dropout probability $p_{\text{uncond}}$, contrastive weight schedule $\lambda(t)$
**Initialize:** Parameters $\theta, \phi$
**for** each mini-batch of size $B$ **do**
    **for** each sample $(\mathbf{x}_0^{(i)}, y^{(i)})$ in mini-batch **do**
        Sample timestep $t \sim \mathcal{U}(\{1, \ldots, T\})$
        Sample noise $\epsilon^{(i)} \sim \mathcal{N}(\mathbf{0}, \mathbf{I})$
        Compute noised inputs: $\mathbf{x}_t^{(i)} = \sqrt{\bar{\alpha}_t}\mathbf{x}_0^{(i)} + \sqrt{1 - \bar{\alpha}_t}\epsilon^{(i)}$
        Drop labels with probability $p_{\text{uncond}}$: $\tilde{y}^{(i)} = \varnothing$ w.p. $p_{\text{uncond}}$
        Predict noise: $\hat{\epsilon}^{(i)} = \epsilon_\theta(\mathbf{x}_t^{(i)}, t, \tilde{y}^{(i)})$
        Extract bottleneck features $\mathbf{h}_t^{(i)}$ of $\mathbf{x}_t^{(i)}$ from U-Net encoder
        Compute projected embeddings: $\mathbf{z}_t^{(i)} = f_\phi(\mathbf{h}_t^{(i)})$
    **end for**
    Compute diffusion loss: $\mathcal{L}_{\text{diff}} = \frac{1}{B} \sum_{i=1}^{B} \|\epsilon^{(i)} - \hat{\epsilon}^{(i)}\|_2^2$
    Compute contrastive loss $\mathcal{L}_{\text{con}}$ using $\{(\mathbf{z}_t^{(i)}, y^{(i)})\}_{i=1}^{B}$ (*e.g.*, using (6) for SupCon)
    Compute total loss: $\mathcal{L}_{\text{CORAL}} = \mathcal{L}_{\text{diff}} + \lambda(t) \cdot \mathcal{L}_{\text{con}}$
    Update $(\theta, \phi)$ using gradients of $\mathcal{L}_{\text{CORAL}}$
**end for**

---

# 4 Experimental Setup and Results

## 4.1 Experimental Setup

**Datasets** We evaluate CORAL on long-tailed (LT) datasets: CIFAR10-LT, CIFAR100-LT [28], CelebA-5 [26], and ImageNet-LT [17]. CIFAR10/100 datasets contain $32 \times 32$ color images broken into 10 and 100 classes. For CIFAR10-LT and CIFAR100-LT, we simulate long-tailed distributions by applying an exponential decay to the class frequencies, controlled by an imbalance factor $\rho \in \{0.01, 0.001\}$. This results in the most frequent (head) class appearing $1/\rho$ times more often than the rarest (tail) class, with intermediate classes following an exponentially decreasing trend. CelebA-5 consists of $64 \times 64$ resized face images in 5 classes corresponding to hair color. CelebA-5 is naturally imbalanced. We construct ImageNet-LT by sampling a subset of ImageNet-2012 following the Pareto distribution with power value $\alpha = 6$. ImageNet-LT has 1,000 classes with class sizes ranging from 5 to 1,280 images, we resize images to $64 \times 64$ resolution. Additional details can be found in Appendix A.

**Implementation** Our implementation builds on the codebase from [5], with modifications to support contrastive latent regularization. We use a U-Net backbone with multi-resolution attention and dropout, consistent across all experiments. We use the SupConLoss implementation from [29]. Training was run on NVIDIA A100 (80 GB SXM) and H100 GPUs. Key training and architectural hyperparameters are summarized in Appendix A.

**Evaluation Metrics** We compute the standard FID [30] and IS [31] to capture both quality and diversity of the generated images. We additionally compute recall for distributions (PRD) and improved recall [32] (labeled as Recall in Table 1). Standard PRD uses $k$-means clustering on InceptionV3 features with 2000 clusters (20 times the number of classes for CIFAR-100) to compute $F_8$ and $F_{1/8}$ scores [33]. $F_8$ emphasizes recall (diversity) and $F_{1/8}$ emphasizes precision (quality). The improved PRD metrics employ $k$-nearest neighbor manifold estimation ($k = 3$) on VGG16

Table 1: Comparison of methods on long-tailed image generation benchmarks.

| Dataset | Method | FID ($\downarrow$) | IS ($\uparrow$) | $F_8$ ($\uparrow$) | Recall ($\uparrow$) | $F_{1/8}$ ($\uparrow$) |
|---|---|---|---|---|---|---|
| CIFAR10-LT ($\rho = 0.01$) $32 \times 32$ | DDPM [2] | 6.17 | 9.43 | 0.87 | 0.52 | 0.94 |
| | CBDM [5] | 5.62 | 9.28 | 0.96 | 0.57 | 0.95 |
| | T2H [6] | 7.01 | 9.63 | 0.89 | 0.54 | 0.95 |
| | CORAL (ours) | **5.32** | **9.69** | **0.97** | **0.59** | **0.97** |
| CIFAR10-LT ($\rho = 0.001$) $32 \times 32$ | DDPM [2] | 13.05 | 9.10 | 0.87 | 0.53 | 0.85 |
| | CBDM [5] | 12.74 | 9.05 | 0.87 | **0.56** | **0.89** |
| | T2H [6] | 12.80 | 8.97 | 0.87 | 0.55 | 0.88 |
| | CORAL (ours) | **11.03** | **9.13** | **0.90** | **0.56** | **0.89** |
| CIFAR100-LT ($\rho = 0.01$) $32 \times 32$ | DDPM [2] | 7.70 | 13.20 | 0.87 | 0.50 | 0.89 |
| | CBDM [5] | 6.02 | 12.92 | 0.91 | 0.56 | 0.90 |
| | T2H [6] | 6.78 | 12.97 | 0.88 | 0.54 | 0.89 |
| | CORAL (ours) | **5.37** | **13.53** | **0.92** | **0.59** | **0.91** |
| CelebA-5 $64 \times 64$ | DDPM [2] | 10.28 | 2.90 | 0.90 | 0.52 | 0.89 |
| | CBDM [5] | 8.74 | 2.74 | 0.92 | 0.57 | 0.90 |
| | T2H [6] | 9.50 | 2.63 | 0.89 | 0.53 | 0.87 |
| | CORAL (ours) | **8.12** | **2.97** | **0.94** | **0.59** | **0.92** |

Table 2: Comparison of methods on ImageNet-LT.

| Dataset | Method | FID ($\downarrow$) | IS ($\uparrow$) | Recall ($\uparrow$) |
|---|---|---|---|---|
| ImageNet-LT $64 \times 64$ | DDPM [2] | 17.08 | 21.03 | 0.39 |
| | CBDM [5] | 22.66 | 17.13 | 0.42 |
| | T2H [6] | 18.59 | 19.15 | 0.44 |
| | CORAL (ours) | **16.11** | **24.17** | **0.48** |

features [34], providing more robust estimates of sample quality and coverage. These metrics collectively provide a comprehensive assessment; in particular, $F_{1/8}$ measures generation fidelity while improved recall and $F_8$ capture the diversity of the generated distribution. FID captures a mixture of both quality and diversity.

For overall metric calculations, we use the balanced version of the datasets for the real data to ensure fair evaluation. All metrics are computed on 50,000 generated samples to ensure statistical reliability. During sampling, class labels are drawn from a uniform distribution across all classes for equal representation.

**Baselines** We compare CORAL's performance against that of DDPM, CBDM, and T2H for the following datasets: CIFAR10-LT with $\rho = 0.01$ and $\rho = 0.001$, CIFAR100-LT, CelebA-5 and ImageNet-LT. We use the publicly available implementations for DDPM [2], CBDM [5], and T2H [6] to train the models with provided parameters, where available, generate synthetic samples, and report results for each of these methods.

## 4.2 Experimental Results

**Comparison of Metrics** In Table 1 and Table 2, we compare the performance of CORAL against standard DDPM, as well as state-of-the-art baselines CBDM and T2H across multiple long-tailed datasets. CORAL consistently outperforms all baselines across all datasets and evaluation metrics, demonstrating its effectiveness in improving both the quality and diversity of generated samples. CORAL demonstrates strongest improvements on metrics that capture diversity and distribution coverage while maintaining improved performance on quality focused metrics

On the large scale ImageNet-LT benchmark with 1,000 classes, CORAL's advantages are most evident, outperforming all baselines by significant margins. Ambient space regularization approaches

exhibit degraded performance scenarios with large numbers of classes, whereas CORAL's latent space intervention maintains consistent performance.

**Per-Class FID**    Figure 3 presents the per-class FID scores for CIFAR10-LT with $\rho = 0.001$, representing a more extreme class imbalance. CORAL consistently outperforms baseline methods across nearly all classes. The gains are particularly notable for the tail classes. Whereas both CBDM and T2H exhibit degraded performance on tail classes, CORAL maintains stable performance across both head and tail classes. For per-class FID analysis in Figure 3, we generate 5K samples for each class and compare against the 5K real samples from the balanced dataset for that specific class.

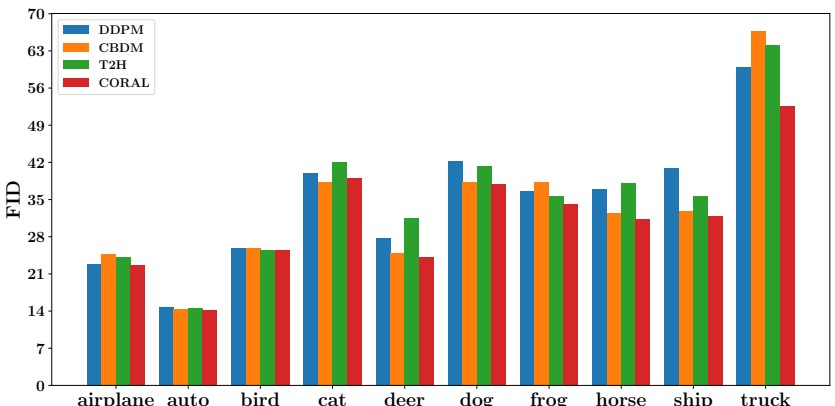

Figure 3: **Per-class FID** ($\downarrow$) for the CIFAR10-LT dataset with an imbalance factor $\rho = 0.001$

**Latent Space Visualizations**    Our experimental results clearly show that CORAL achieves better performance by explicitly enforcing class-wise separation in the latent space. In Figure 1, we present t-SNE visualizations of U-Net bottleneck representations for CIFAR10-LT with an imbalance ratio of $\rho = 0.01$, comparing models trained with DDPM (on both the original balanced CIFAR-10 and CIFAR10-LT) and with CORAL on CIFAR10-LT. Figure 2 visualizes the separated representations (using t-SNE) learned by CORAL for the CelebA-5 dataset. Additional visualizations for the other datasets using both t-SNE and UMAP [35] are included in Appendix D. In particular, our plots for a balanced dataset with a limited number of samples per class show that the observed representation entanglement arises predominantly from class imbalance in the training distribution.

**Generation Quality**    Figure 4 presents generated samples from CBDM, T2H, and CORAL for the `tulips` class (class 92) in CIFAR100-LT. Visually, CORAL produces samples that are both more diverse and of higher fidelity compared to the other methods. These qualitative differences align with the quantitative improvements observed in Table 1, where CORAL achieves superior performance across all evaluated metrics. CBDM suffers from mode collapse by producing smaller flowers with excessive grass backgrounds borrowed from head animal classes. T2H shows diminished class fidelity as tulips resemble other flower types due to over-transfer from head to tail classes. In contrast, CORAL generates tulips that reflect appropriate scale and structure, with distinctive features and backgrounds consistent with the training data. This demonstrates CORALs ability to balance the trade-off between preserving tail-class characteristics and promoting sample diversity. Additional visualizations are provided in Appendix B.

**Ablation Studies**    We have performed extensive ablation studies for various hyperparameters, including the SupCon temperature $\tau_{\text{SC}}$, the time-dependent weighting function temperature $\tau_r$, and the CFG sampling parameter $\omega$; these plots can be found in Appendix C.

## 5   Concluding Remarks

**Broader Impacts**    As generative models have become more widely utilized in practice, their representativeness becomes more impactful. Tail class generation has become a key method to address long-tailed recognition tasks such as disease detection where real data is limited. While

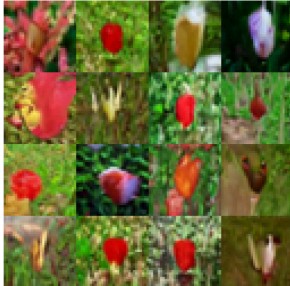 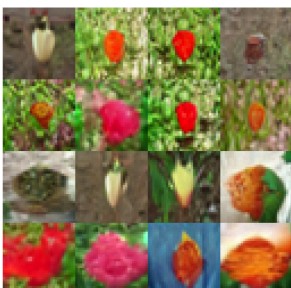 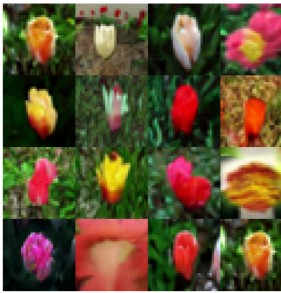

Figure 4: **Comparison of generated samples** from the class `tulips` (class 92) in CIFAR100-LT, $\rho = 0.01$. CBDM (left), T2H (middle), and CORAL (right). CORAL shows increased diversity and fidelity relative to existing approaches.

generated images have the potential to cause harm, *e.g.* deepfakes or bias amplification, CORAL helps to mitigate the bias introduced by dataset imbalance.

**Future Directions**  While our work focuses on class conditional generation with categorical imbalance, the core principle of latent space disentanglement through contrastive regularization extends naturally to more complex generative settings. A particularly promising application domain is the fine tuning of large scale text-to-image (T2I) models for specialized tasks. When pretrained models like Stable Diffusion [36] are adapted to domain specific applications, such as medical imaging, scientific visualization, or specialized industrial use cases, the fine tuning datasets often exhibit severe imbalance. For T2I models that employ U-Net architectures, CORAL's approach of applying contrastive regularization at these bottleneck layers could prevent rare concepts from becoming entangled with common ones. Exploring CORAL's applicability to parameter efficient fine tuning methods, *e.g.* LoRA [37], for domain adaptation and investigating whether similar contrastive interventions can improve generation quality for underrepresented concepts constitute promising directions for ensuring equitable representation across specialized domains.

**Limitations**  While CORAL is able to produce diverse and high quality images when trained on heavily imbalanced datasets, its power comes at the cost of additional computational complexity. This limitation is shared by all comparable methods, though it can be reduced by finetuning with the CORAL loss rather than fully training.

**Conclusions**  Ensuring high-quality sample generation for tail classes of long-tailed datasets remains a major challenge. In addressing this challenge, we have revealed a previously unknown cause for the poor performance of DMs: the (U-Net) latent representations for the tail classes completely overlap with those for the head classes, thereby severely limiting the guidance of the former. Our method, CORAL, significantly enhances both the diversity and fidelity of diffusion model outputs relative to the state-of-the-art by separating and realigning the latent space representations, especially for the long-tail classes using contrastive losses. We have demonstrated that CORAL performs well for datasets with both extreme imbalance and many classes, and our results suggest that disentangling in the latent space is more effective than rebalancing and increased guidance in the ambient space.

## Acknowledgments and Disclosure of Funding

This work is supported in part by NSF grants CIF-1815361, CIF-2007688, DMS-2134256, and SCH-2205080.

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

# A  Additional Dataset and Experimental Setup Details

**Long-Tail Datasets**  Balanced datasets can be artificially transformed into long-tail (LT) datasets by assigning class sample counts according to a geometric progression governed by an imbalance ratio $\rho$. In this formulation, the head class has the maximum number of samples, $N$, while the tail class has approximately $\rho N$ samples. The number of samples in class $i$ is given by:

$$n_i = \left\lfloor N\rho^{\frac{i}{C-1}} \right\rfloor \tag{8}$$

where $N$ is the number of samples in the head class, $\rho$ is the imbalance ratio ($0 < \rho < 1$), $i \in \{0, 1, \ldots, C-1\}$ is the class index, and $C$ is the total number of classes.

**CIFAR10-LT and CIFAR100-LT**  The original CIFAR10 and CIFAR100 datasets each consist of a training set with 50k images uniformly distributed across 10 or 100 classes, respectively. Their long-tailed variants, CIFAR10-LT and CIFAR100-LT [28], introduce an exponential decay in class frequency from class 0 to the final class. Common long-tail imbalance ratios include $\rho = 0.01$ and $\rho = 0.001$. Specifically for $\rho = 0.01$, CIFAR10-LT contains 12,406 images, with the first head class comprising 5,000 samples and the last tail class only 50. CIFAR100-LT has 10,847 images, with the head class containing 500 samples and the tail just 5.

Experiments on CIFAR10-LT and CIFAR100-LT were conducted using NVIDIA A100 80GB SXM GPUs. Training took approximately 7 hours, and sampling required 8 hours. For DDPM, CBDM, and CORAL, the hyperparameters used were: a learning rate of $2 \times 10^{-4}$, batch size of 128, Adam optimizer with default momentum parameters, dropout rate of 0.1, 150k training steps, and $T = 1000$ diffusion steps. For T2H, all settings remained the same except for the number of training steps, which was increased to 200k.

**CelebA-5**  CelebA-5 [16] is a five-class subset of the CelebA dataset, composed of samples labeled with exactly one of the following hair colors: black, brown, blonde, gray, or bald. Samples with multiple or missing labels are excluded. The dataset is naturally imbalanced, with black- and brown-haired individuals significantly outnumbering those with gray hair or baldness.

Experiments on CelebA-5 were run on NVIDIA H100 GPUs. Training took approximately 18 hours, and sampling required 22 hours. All models were trained with a learning rate of $3 \times 10^{-4}$ and a batch size of 128, with all remaining hyperparameters kept consistent with the CIFAR experiments.

**ImageNet-LT**  ImageNet-LT is a long-tailed variant of ImageNet-2012 constructed by [17] by sampling a subset following the Pareto distribution with power value $\alpha = 6$. The dataset comprises 115.8k images from 1000 categories, with a maximum of 1,280 images per class and a minimum of 5 images per class.

Experiments on ImageNet-LT were conducted using NVIDIA H100 GPUs. All models were trained with a batch size of 128, 300k training steps, and $T = 1000$ diffusion steps. For evaluation, we generated 50k samples uniformly across all classes and compared against the balanced validation set containing 20k images as the real samples.

**Hyperparameters**  Table 3 summarizes the regularization hyperparameters and sampling guidance scale $\omega$ used for each method and dataset. The sub-tables correspond to DDPM (top left), CBDM (top right), T2H (bottom left), and CORAL (bottom right), respectively. For CORAL, the base contrastive weight, $w$, in (7) was set to $0.01$.

We follow the code implementation for CBDM [5] with the regularization weight $\tau_{\text{cb}} = \tau/T$, where $\tau$ is the original weight defined in [5], and $T$ is the total number of diffusion timesteps. For T2H [6], we use the code implementation with direct distance-based weighting that does not require a regularization parameter.

Table 3: Hyperparameter settings for each method.

**DDPM [2]**

| Dataset | $\omega$ |
|---|---|
| CIFAR10-LT ($\rho = 0.01$) | 0.8 |
| CIFAR10-LT ($\rho = 0.001$) | 1.0 |
| CIFAR100-LT ($\rho = 0.01$) | 0.8 |
| CelebA-5 | 0.6 |

**CBDM [5]**

| Dataset | $\omega$ | $\tau_{\mathrm{cb}}$ |
|---|---|---|
| CIFAR10-LT ($\rho = 0.01$) | 1.0 | 1.0 |
| CIFAR10-LT ($\rho = 0.001$) | 1.8 | 1.0 |
| CIFAR100-LT ($\rho = 0.01$) | 1.6 | 1.0 |
| CelebA-5 | 1.0 | 50.0 |

**T2H [6]**

| Dataset | $\omega$ |
|---|---|
| CIFAR10-LT ($\rho = 0.01$) | 1.0 |
| CIFAR10-LT ($\rho = 0.001$) | 1.7 |
| CIFAR100-LT ($\rho = 0.01$) | 1.5 |
| CelebA-5 | 1.0 |

**CORAL (ours)**

| Dataset | $\omega$ | $\tau_{\mathrm{SC}}$ | $\tau_r$ |
|---|---|---|---|
| CIFAR10-LT ($\rho = 0.01$) | 0.6 | 0.12 | 0.8 |
| CIFAR10-LT ($\rho = 0.001$) | 1.0 | 0.10 | 1.0 |
| CIFAR100-LT ($\rho = 0.01$) | 0.8 | 0.09 | 1.0 |
| CelebA-5 | 0.7 | 0.12 | 0.8 |

# B  Additional Results

**Comparing Contrastive Regularization in Ambient and Latent Space**  CORAL differs fundamentally from ambient space contrastive regularization methods (similar to DiffROP [4]) in both its approach and effectiveness. CORAL operates directly within the diffusion model's internal latent space, specifically at the U-Net bottleneck layer augmented with a projection head, where semantic representations are formed and class-discriminative embeddings are learned. This architectural difference is crucial because CORAL addresses the representation entanglement where tail-class samples overlap heavily with head-class representations. In contrast, ambient space contrastive regularization methods enforce separation constraints on the image space.

| Dataset | Method | FID ($\downarrow$) | IS ($\uparrow$) | Recall ($\uparrow$) |
|---|---|---|---|---|
| CIFAR10-LT ($\rho = 0.01$) | DDPM [2] | 6.17 | 9.43 | 0.52 |
| | Ambient Space Contrastive | 5.85 | 9.18 | 0.55 |
| | CORAL (ours) | **5.32** | **9.69** | **0.59** |
| ImageNet-LT $64 \times 64$ | DDPM [2] | 17.08 | 21.03 | 0.39 |
| | Ambient Space Contrastive | 24.73 | 15.12 | 0.34 |
| | CORAL (ours) | **16.11** | **24.17** | **0.48** |

Table 4: Comparison of contrastive regularization strategies on CIFAR10-LT and ImageNet-LT.

Table 4 provides empirical validation of CORAL's design choices. The results show that learned separation in the latent space scales more effectively than imposed separation in the ambient space. As the generation task becomes more challenging, ambient space methods not only struggle but actually degrade below baseline DDPM performance in distributional coverage as the class space grows, whereas CORAL maintains its effectiveness across all metrics.

**Generated Images**  We generate images for CIFAR10-LT, CIFAR100-LT, and CelebA-5 to evaluate the effectiveness of CORAL in addressing the challenges of diffusion models trained on long-tailed datasets. Randomly selected examples are shown in Figures 5 to 7, illustrating how our contrastive latent alignment framework improves both the quality and diversity of generated samples, particularly for tail classes.

Figures 5 and 6 show generated samples for CIFAR10-LT and CIFAR100-LT, respectively, with $\rho = 0.01$. CORAL successfully disentangles latent representations to generate high-quality, diverse samples for the underrepresented classes. CORAL preserves the distinctive characteristics of each

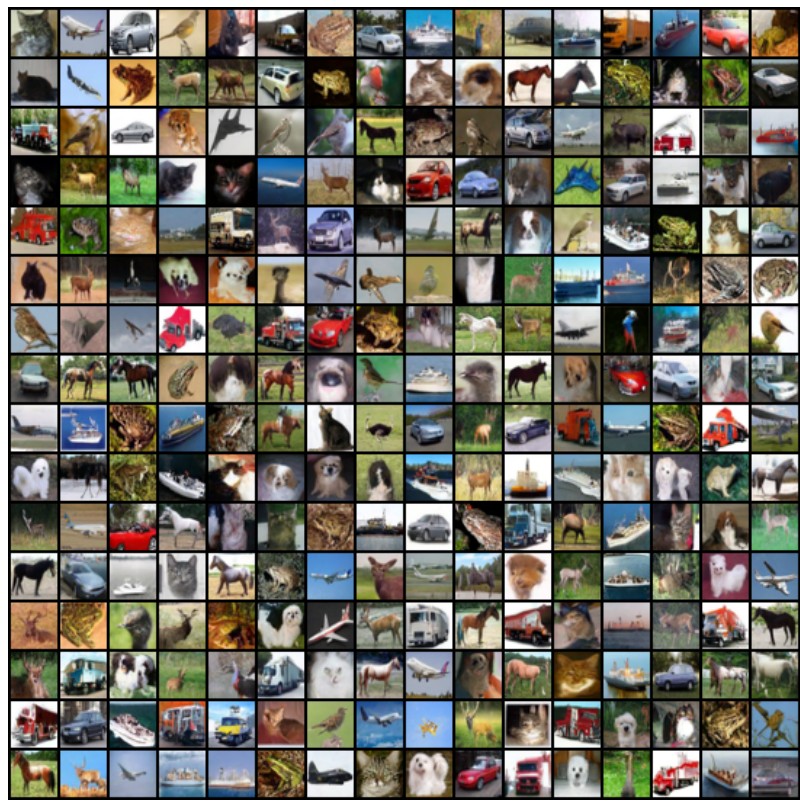

Figure 5: **Generated samples** produced by CORAL on the CIFAR10-LT dataset with $\rho = 0.01$.

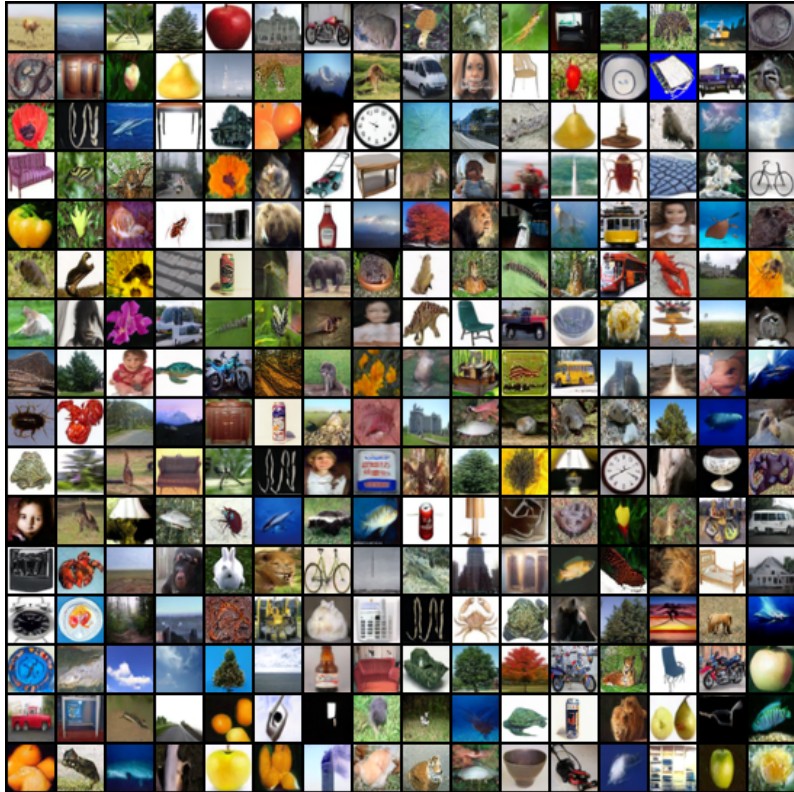

Figure 6: **Generated samples** produced by CORAL on the CIFAR100-LT dataset with $\rho = 0.01$.

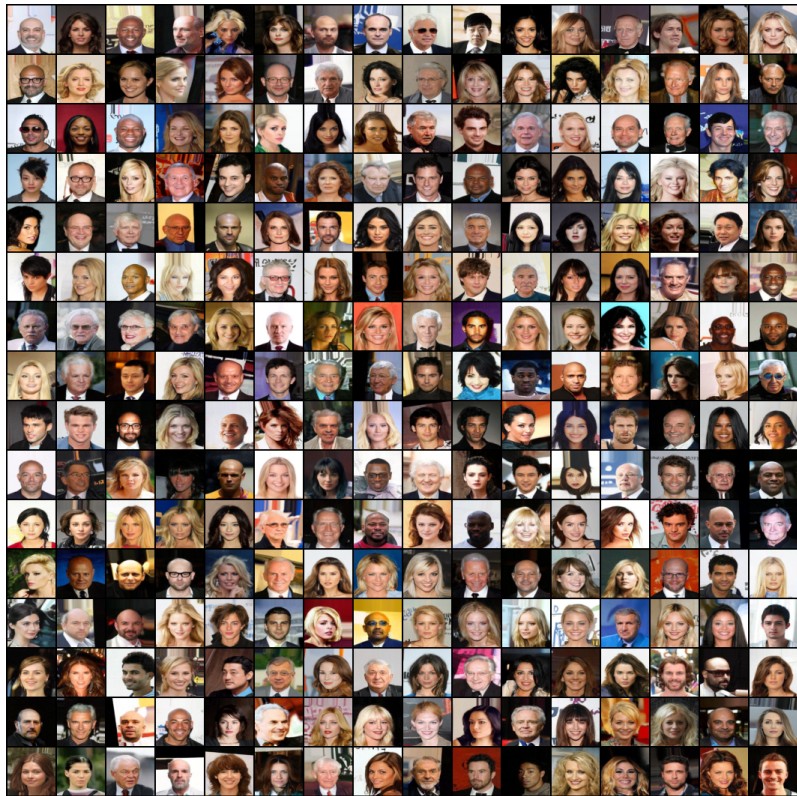

Figure 7: **Generated samples** produced by CORAL on the CelebA-5 dataset.

class, effectively mitigating feature borrowing from head to tail classes. The visual quality of these results highlights the effectiveness of CORALs latent space regularization in promoting class separation and maintaining clean, well-structured features in the generated outputs.

Figure 7 displays generated samples for CelebA-5, demonstrating CORALs ability to handle naturally imbalanced data. The dataset exhibits pronounced class imbalance across five hair color categories (black, brown, blond, gray, and bald) with the head class containing nearly 15 times more samples than the tail class. In such imbalanced settings, latent representations for tail classes often become entangled with those of head classes. CORAL effectively preserves class-specific features, producing diverse and realistic images in all categories.

## C   Ablation Studies

**Effects of Hyperparameters**   Figure 8 illustrates the effect of three key hyperparameters on CORALs performance for CIFAR10-LT with imbalance ratio $\rho = 0.01$, measured by FID. The supervised contrastive temperature, $\tau_{SC}$, in (5) achieves optimal performance at $\tau_{SC} = 0.12$, beyond which FID increases sharply.

For the decay rate temperature, $\tau_r$, in (7), the best performance is observed at $\tau_r = 0.8$. FID remains relatively stable for $0.7 \leq \tau_r \leq 0.9$, with a steep increase outside of this range. These findings support our hypothesis that contrastive regularization is most effective when applied toward the end of the denoising process (i.e., when $t \sim 0$).

Finally, for the CFG scale, $\omega$, in (4), FID traces a convex curve with optimal performance at $\omega = 0.6$. This indicates a trade-off between leveraging class-conditional information ($\omega > 0$) and avoiding over-conditioning that could limit sample diversity ($\omega \gg 0.6$). These results underscore the importance of careful hyperparameter tuning in CORAL to achieve an optimal balance between sample fidelity and diversity, particularly for long-tailed datasets.

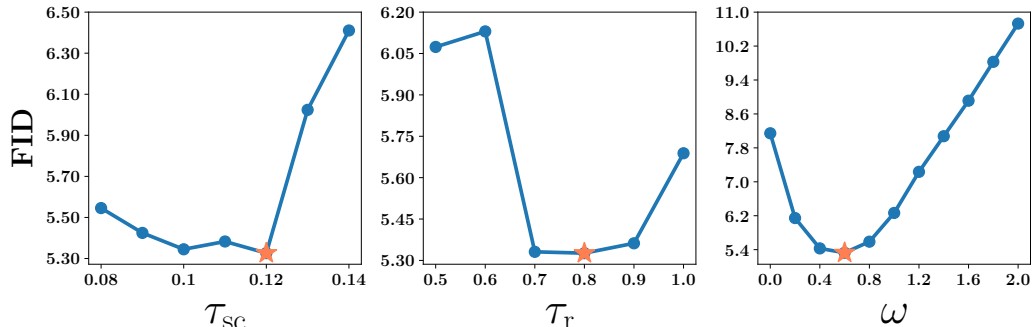

Figure 8: **Effect of regularization hyperparameters and guidance scales on FID.** From left to right: FID vs $\tau_{SC}$, FID vs $\tau_r$, and FID vs $\omega$ for the CIFAR10-LT dataset. Coral stars mark the lowest FID achieved for each hyperparameter.

**CORAL on Balanced Datasets**    We also evaluate CORAL on two balanced datasets, CIFAR10 and CIFAR100, using FID as the performance metric, as shown in Table 5. Even in the absence of class imbalance, CORAL outperforms DDPM and CBDM in terms of FID. This improvement stems from CORALs contrastive loss, which promotes class-wise separation in the latent space even in balanced settings, as illustrated in Figure 9. Importantly, this separation is achieved without compromising fidelity or diversity, as reflected in the consistently strong FID scores.

Table 5: Comparison of methods on CIFAR10 and CIFAR100 image generation.

| Dataset | Method | FID ($\downarrow$) |
|---------|--------|--------|
| CIFAR10 | DDPM [2] | 3.84 |
|  | CBDM [5] | 3.61 |
|  | CORAL (ours) | **3.30** |
| CIFAR100 | DDPM [2] | 3.91 |
|  | CBDM [5] | 3.37 |
|  | CORAL (ours) | **2.86** |

**Impact of Sample Size in Balanced Datasets**    Figure 10 illustrates the impact of total sample size on latent representations from DDPM for the balanced CIFAR10 dataset. As the number of training samples per class decreases from 5k to 100, we observe a noticeable reduction in cluster formation in the latent space, with representations becoming increasingly scattered. This suggests that in such a highly overparameterized regime, the model memorizes individual samples rather than learning generalizable class-level structure. We note that this type of scattering is visually distinct from the entanglement observed in DDPM under class imbalance (see, for example, Figure 12). For imbalanced datasets, models have more available information on head than on tail classes, allowing them to learn better representations for the former than the latter. This does not lead to scattering, but rather to a distinct overlap of tail class representation clusters within head class clusters.

Figure 11 illustrates the memorization behavior of DDPM on the balanced CIFAR10 dataset with 50 samples per class and guidance strength $\omega = 0.1$. For each class, the panel displays two rows of 10 images: the bottom row (outlined in magenta) shows samples generated by DDPM, while the top row presents the most visually similar real training samples corresponding to each generated image. The generated samples are near-identical to the training samples, differing at most by a horizontal reflection, highlighting the extent of memorization in this limited-data setting.

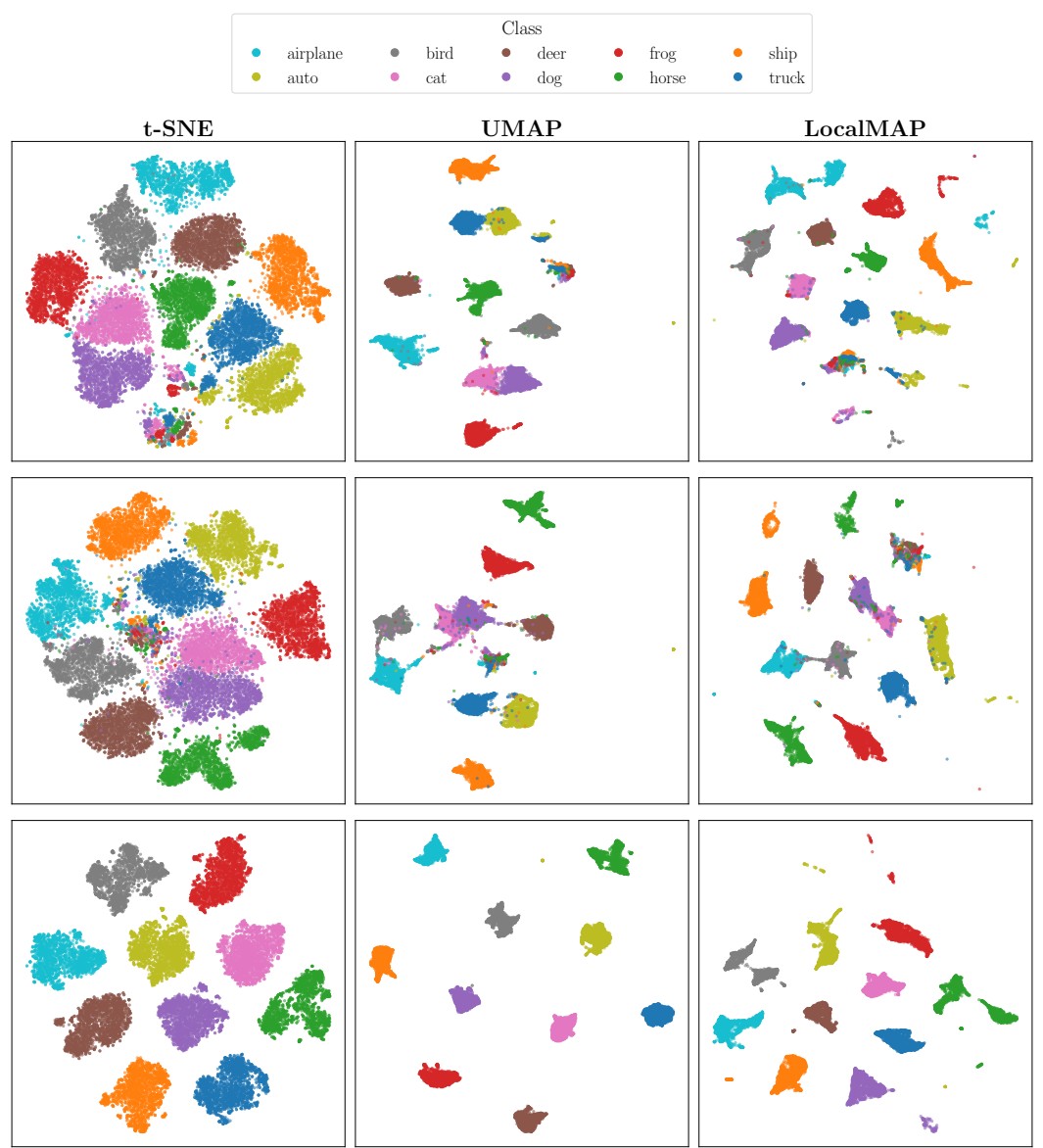

Figure 9: **Visualizations of U-Net bottleneck features** using t-SNE, UMAP, and LocalMAP on CIFAR-10 (balanced). Each row shows a different method trained on CIFAR-10, listed **from top to bottom**: DDPM [18], CBDM [5], and CORAL (ours).

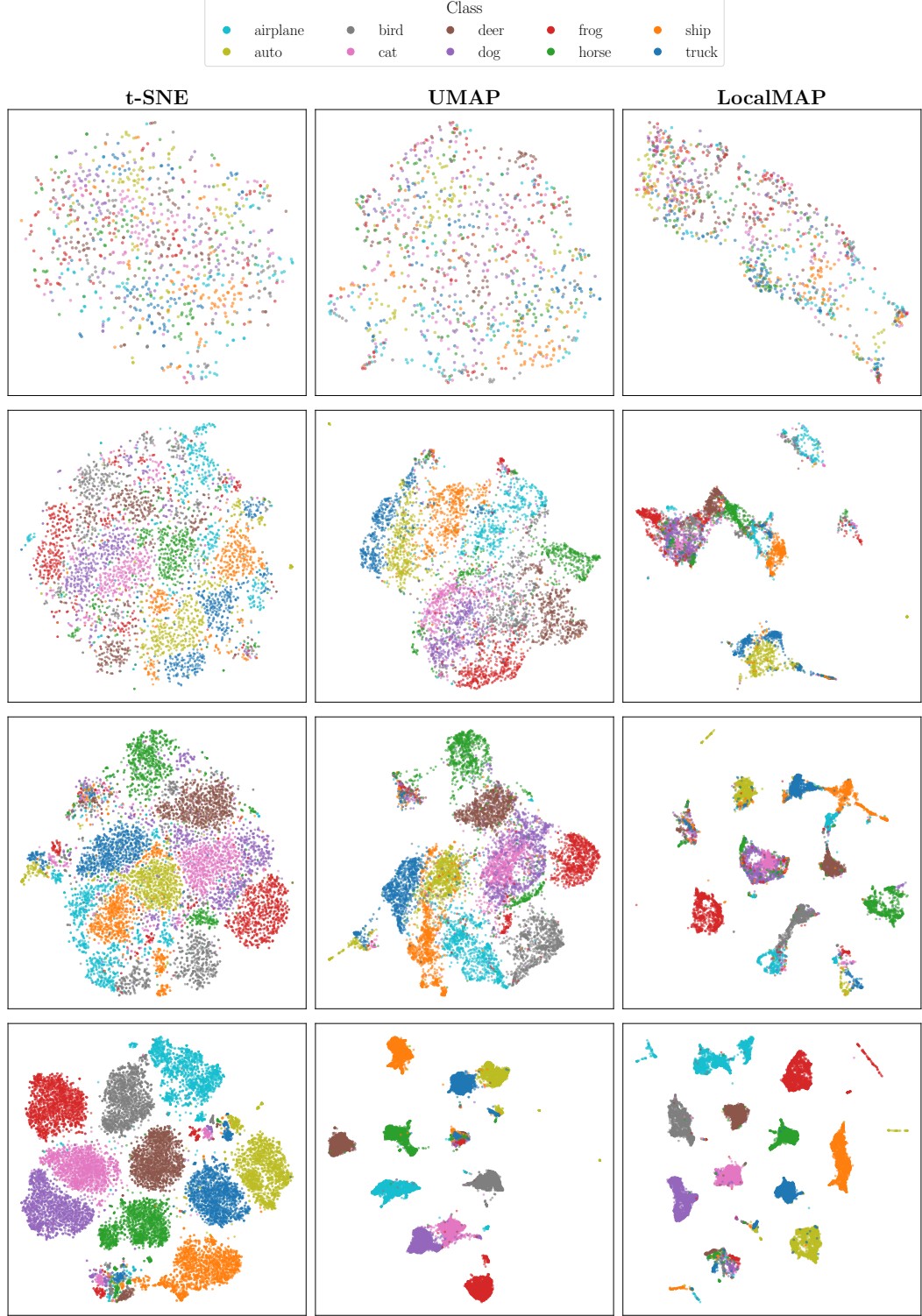

Figure 10: **Visualizations of U-Net bottleneck features** using t-SNE, UMAP, and LocalMAP on **CIFAR10 (balanced) for DDPM** models trained with varying amounts of data. Each row corresponds to a different trained model for DDPM , **with increasing samples per class from top to bottom: 100, 500, 1k, and 5k**. For the final row (5k samples per class), a randomly selected subset of 20k samples is visualized from the full training set of 50k.

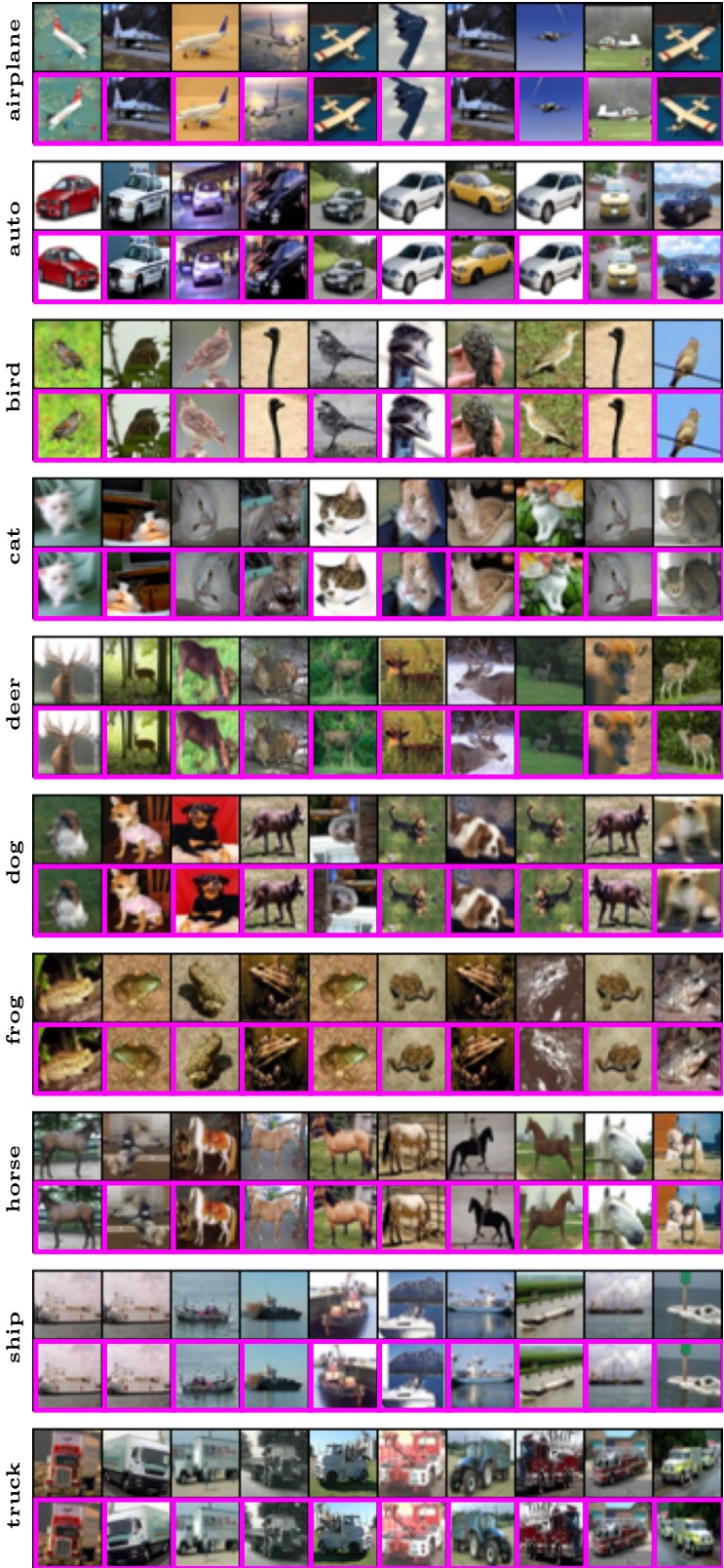

Figure 11: **Memorization in limited-data scenarios.** For each class, the top row shows the closest matching real training samples corresponding to the generated images in the bottom row (outlined in magenta), which were produced by a DDPM model trained on CIFAR-10 with 50 samples per class and sampled using $\omega = 0.1$. The high visual similarity between generated and training samples reflects the models strong memorization behavior in limited-data settings.

# D Comparison with Other Methods

**Latent Space Visualization**    Figure 12 visualizes the latent representations from the U-Net bottleneck layer using t-SNE [11] (left), UMAP [35] (middle), and LocalMAP [38] (right) for DDPM [18], CBDM [5], T2H [6], and CORAL trained on CIFAR10-LT with an imbalance ratio of $\rho = 0.01$. CORAL exhibits markedly improved class-wise separation in latent space, mitigating the representational entanglement that typically causes feature mixing between head and tail classes. Figure 13 presents analogous visualizations for CelebA-5 across all methods except T2H, for which no implementation is available on this dataset.

**Comparison with Baseline Methods**    We compare CORAL with baseline methods and highlight its strengths to address long-tailed generation in diffusion models.

- **CBDM:** Introduces a distribution adjustment regularizer during training that encourages similarity between generated images across different classes, transferring knowledge from head classes to tail classes. CBDM [5] suffers from mode collapse because its regularization loss encourages the model to produce similar outputs across different class conditions.

- **T2H:** Employs weighted denoising score matching to transfer knowledge from head classes to tail classes by using head samples as denoising targets for noisy tail samples. Its performance depends on both label distribution and sample similarity. T2H's [6] score substitution mechanism could potentially lead to mode collapse when noisy tail samples are consistently mapped to the same limited set of head references due to similarity-based selection.

- **CORAL:** As demonstrated in our experimental results, CORAL consistently outperforms both CBDM and T2H across a range of evaluation metrics, with particularly strong gains in tail classes. Our experimental results across multiple datasets highlight the strengths of CORAL:

  1. **Mode Stability**: CORAL prevents mode collapse, and generates class-consistent and visually diverse samples. As can be seen in the generated samples. This is in contrast to CBDM, which often fails to preserve class identity, *e.g.,* by generating class-conditioned samples displaying attributes of other classes, as shown in Figure 4.
  CORAL effectively reduces undesirable interclass feature borrowing in the class labeled datasets we consider. At the same time, CORAL allows the transfer of non-discriminative features that facilitate generalization, as shown in Figure 4.

  2. **Adaptive Regularization**: CORAL incorporates time-dependent regularization into the contrastive loss. This adaptive weighting enhances separation during the later stages of denoising, when outputs are less noisy and more semantically meaningful. Figure 8 shows that contrastive regularization is most effective when applied toward the end of the denoising process.

  3. **Latent Disentanglement**: CORALs strength lies in leveraging the lower-dimensional latent space of the denoising U-Net, which has been shown to capture semantically meaningful structure [10]. CORAL achieves effective inter-class disentanglement in the latent space by employing a linear projection head (see Figure 2), resulting in high-fidelity and class-aligned generated samples. These effects are illustrated in Figure 12 for CIFAR10-LT with $\rho = 0.01$ and in Figure 13 for CelebA-5, using latent space visualizations from t-SNE, UMAP, and LocalMAP.

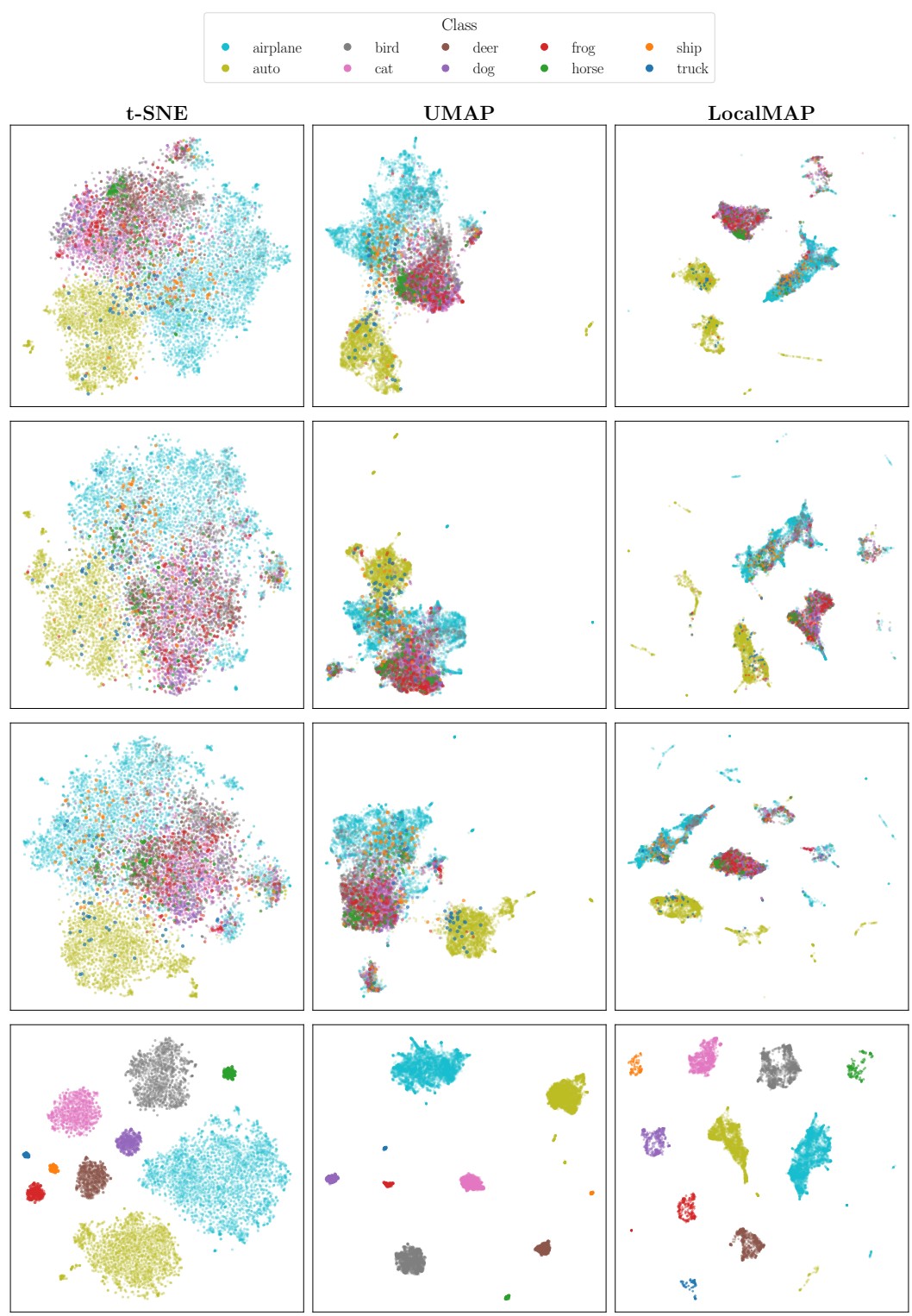

Figure 12: **Visualizations of U-Net bottleneck features** using t-SNE, UMAP, and LocalMAP on CIFAR10-LT with an imbalance ratio of $\rho = 0.01$. Each row shows a different method trained on CIFAR10-LT, listed **from top to bottom**: DDPM [18], CBDM [5], T2H [6], and CORAL.

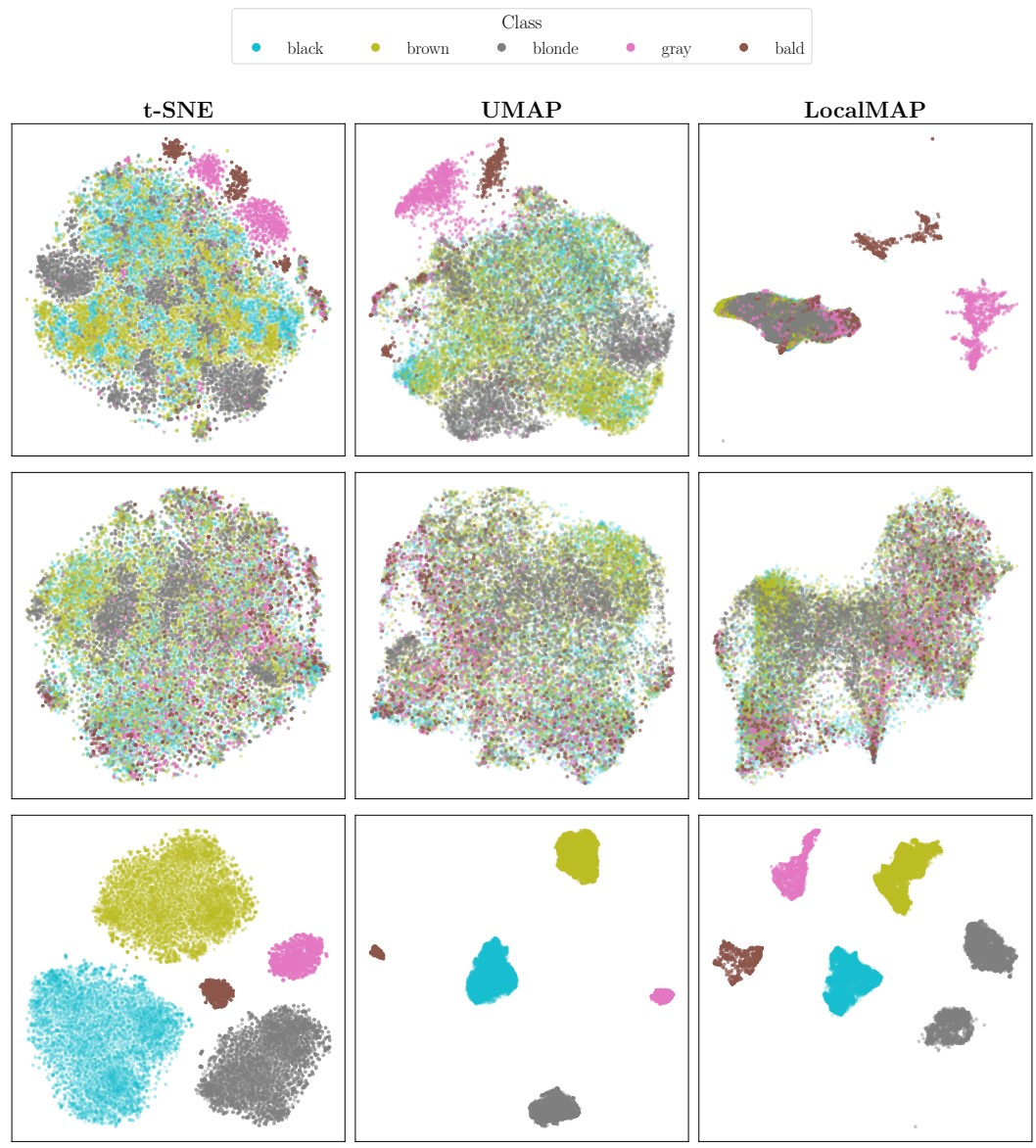

Figure 13: **Visualizations of U-Net bottleneck features** using t-SNE, UMAP, and LocalMAP on CelebA-5. Each row shows a different method trained on CelebA-5, listed **from top to bottom**: DDPM [18], CBDM [5], and CORAL.

