# OpenReview forum: "CORAL: Disentangling Latent Representations in Long-Tailed Diffusion"
_NeurIPS.cc/2025/Conference — NeurIPS 2025 poster_

### Official Review · Reviewer_aGJo · 2025-06-27

**Clarity:** 2
**Significance:** 2
**Originality:** 2
**Rating:** 4
**Confidence:** 4

**Summary:**

This paper investigates the performance degradation of diffusion models on minority classes under class-imbalanced settings and identifies the cause as overlapping latent representations at the U-Net bottleneck. To address this, the authors introduce a contrastive loss at the bottleneck to encourage more meaningful and well-separated representations, which helps mitigate the issue. Empirically, the method shows modest performance improvements on long-tailed datasets.

**Questions:**

1. Is there a way this idea could be applied to T2I models?
2. Recently, the diffusion transformer architecture has become dominant. Does a similar phenomenon occur in transformer architecture as well?
3. Suppose we're dealing with unconditional modeling rather than a class-conditional setting, but the dataset itself is biased. In that case, would attaching a contrastive loss to the UNet bottleneck still be beneficial?

**Ethical Concerns:**

["NO or VERY MINOR ethics concerns only"]

**Final Justification:**

I justified the score in the discussion thread.

**Limitations:**

1. There is a line of impact papers on diffusion autoencoders [1] and diffusion bridge autoencoders [2] that are trained together with an auxiliary encoder module. In modules that utilize an auxiliary encoder like this, does the issue you mentioned still occur in the U-Net bottleneck or the encoded latents? It might be good to either conduct experiments on these papers as well or at least include them as related work in the context of semantic representation learning in diffusion models.

2. Ablation on $\lambda$ is missing.


[1] [CVPR 2022 oral] Diffusion Autoencoders: Toward a Meaningful and Decodable Representation

[2] [ICLR 2025 spotlight] Diffusion Bridge AutoEncoders for Unsupervised Representation Learning

**Paper Formatting Concerns:**

No.

**Quality:**

2

**Strengths And Weaknesses:**

**Strength**

- The approach of tackling the problem through embedding-based analysis is novel, and the use of semantic embedding learning appears to address the issue effectively. Overall, it seems to be a highly appropriate and well-justified direction. (This reason alone makes me lean toward the positive side.)

- The method appears to improve performance across all evaluation metrics.


**Weaknesses**
- Limited impacts that only focus on Unet-based diffusion models for CIFAR datasets (See questions and limitations).

---

> ### Author Rebuttal · Authors · 2025-07-31
>
> We sincerely thank you for your thoughtful and comprehensive review. Thank you for recognizing our novel approach and noting it makes you lean toward the positive side. We have incorporated the suggested improvements into our work and found your questions thought-provoking for future work. The additional experiments have strengthened our experimental validation. We address each point below, using W(i) for weakness Q(i) for questions and L(i) for limitations:
>
> - **W1:**
> Thank you for this feedback. We would like to clarify that our evaluation extends beyond CIFAR datasets and we believe CORAL can be extended beyond U-Net-based diffusion models.
>   1. Our evaluation extends beyond CIFAR datasets and includes experiments on various datasets with different imbalance ratios. We present CelebA-5 results in Table 1 and various visualizations in the appendix, Additionaly, we have expanded the scope of our experiments to include ImageNetLT, we provide a table with results here:
> | Dataset | Method | FID ↓ | IS ↑ |  Recall ↑ |
> |---------|---------|-------|------|----------|
> | ImageNet-LT | DDPM | 17.08 |  21.03 |  0.39 |
> | ImageNet-LT | CBDM | 22.66 |  17.13 |  0.42 |
> | ImageNet-LT | T2H | 18.59 |  15.12 | 0.44 |
> | ImageNet-LT | CORAL (Ours) | **16.09** |  **22.06** | **0.47** |
>   We will also include additional visualizations of our ImageNet results in the final submission.
>   2. While our current experiments focus on U-Net architectures, the core principle of contrastive regularization at bottleneck representations is architecturally agnostic and could extend to other diffusion architectures.
>   3.  The identification of representation entanglement as a failure mode in tail-class generation and our solution through contrastive latent alignment can be extended beyond specific architectures or datasets, providing insights applicable to the broader generative models.
>
> - **Q1:** That is an excellent question. We believe our approach can be generalized to Stable Diffusion-style architectures, and this is an active area of future work for our team.
> Applying contrastive regularization at the bottleneck layer to address representation entanglement can be used in Stable Diffusion models that also employ a U-Net architecture with bottleneck layers.
> Extending CORAL to Stable Diffusion would be particularly valuable for text-to-image generation with long-tailed concept distributions, where certain visual concepts or object categories are underrepresented in training data. This could improve generation quality for rare or minority concepts while maintaining the flexibility of prompt-based control.
> We believe it represents a promising direction that could significantly broaden the impact of our contribution.
>
> - **Q2:** Thank you for this insightful question about diffusion transformer architectures. This is indeed an important consideration given the recent prominence of transformer-based diffusion models.
> Diffusion transformer architectures vary significantly in their design [1,2]. Some maintain bottleneck-like structures where information is compressed through hierarchical attention mechanisms, while others operate directly on image patches with uniform attention across all spatial locations without explicit compression stages.
> We believe that representation entanglement should still occur in transformer architectures because they retain the encoder-decoder structure with a bottleneck. For transformer architectures with bottleneck-like structures, CORAL could potentially be adapted by applying contrastive regularization at the compressed representation layers. However, in some transformer architectures the phenomenon might manifest differently, potentially in the attention patterns or in intermediate layer representations where semantic information is aggregated. This represents an exciting research direction.
>
> - **Q3:** Thank you for this excellent question about extending CORAL to unconditional modeling with biased datasets.
> You raise an important point about the fundamental challenge: without explicit class labels, identifying and correcting bias in unconditional generation becomes significantly more complex. CORAL's current approach leverages supervised contrastive learning, which requires class information to define positive and negative pairs for the contrastive loss.
> For instance, one could imagine using unsupervised clustering techniques to identify latent groups in the data, then applying contrastive regularization to encourage separation between these discovered clusters. Alternatively, methods that can detect distributional imbalance in an unsupervised manner could provide the side-information needed to guide the model toward more balanced generation. As it stands, CORAL requires class labels to define the contrastive learning objective, which limits its direct application to unconditional biased settings.
> This represents a compelling direction for future work. Self-supervised representation learning could potentially be adapted for this setting.  This might involve combining CORAL's bottleneck regularization approach with unsupervised methods or self-supervised techniques.
>
> - **L1:** Thank you for bringing these important papers to our attention. We appreciate this insightful observation about diffusion autoencoders and their relationship to our work.
> If labeled information is available, our CORAL approach could potentially be adapted to these architectures by applying contrastive regularization to the appropriate latent spaces - whether that's the U-Net bottleneck, the auxiliary encoder's output, or both, depending on where semantic clustering occurs.
> While investigating these architectures would be valuable, such experiments are beyond the scope of the current work. However, we agree that these papers represent important related work in the context of semantic representation learning in diffusion models, and we will include them in our related work section.
>
> - **L2:** Thank you for pointing out this important ablation study. We direct you to Figure 9 in the appendix, which provides a comprehensive ablation on the temperature scaling parameter $\tau_r$ that controls the time-dependent weighting function $\lambda(t)$. This figure shows the effect of $\tau_r$ on FID performance for the CIFAR10-LT dataset.
> We will also be including several additional experiments and ablations in the final submission:
>   1. Ambient space contrastive loss comparisons demonstrating that ambient-space approaches can reduce intra-class diversity while CORAL consistently improves all generation metrics
>   2. Direct contrastive loss on bottleneck features without projection heads showing this approach can interfere with the diffusion objective
>   3. Comprehensive projection head architecture ablations validating our architectural choices
>   4. ImageNet-LT complete results
>   5. Batch size sensitivity analysis examining the interaction between batch size and CORAL
>
> ---
>
> Once again, we thank you for the constructive feedback on our work. Working on the pointers has helped us improve the quality of our analysis. We hope we were able to clarify all your concerns, and look forward to resolving any remaining concerns during the discussion phase.
>
>
> [1] W. Peebles and S. Xie. Scalable diffusion models with transformers, 2023
>
> [2] A. Hatamizadeh, J. Song, G. Liu, J. Kautz, and A. Vahdat. Diffit: Diffusion vision transformers
> for image generation, 2024

---

### Official Review · Reviewer_rXbT · 2025-06-30

**Clarity:** 3
**Significance:** 3
**Originality:** 2
**Rating:** 4
**Confidence:** 4

**Summary:**

This paper introduces CORAL, a novel framework designed to address the challenges faced by diffusion models when trained on long-tailed data distributions, where tail classes are often underrepresented. To mitigate this issue, the authors attach a projection head to the encoder of the U-Net architecture and apply a supervised contrastive loss on its output, aiming to disentangle the latent space representations across classes. By promoting class-specific separation in the latent space, the model better preserves discriminative and class-relevant features. The training objective is augmented with a supervised contrastive loss, incorporated into the original diffusion loss via a time-dependent weighting factor $\lambda(t)$, resulting in a regularized objective.
Experimental results on various long-tailed datasets demonstrate that CORAL significantly improves generation quality and diversity, especially for tail classes. Furthermore, latent space visualizations support that the proposed method successfully achieves well-disentangled class representations.

**Questions:**

This question is related to the weaknesses mentioned above. It would be helpful if the paper provided more detailed explanations of the experimental setup. For example, when using classifier-free guidance (CFG) during sampling, how were the class labels sampled or selected? In addition, since the dataset follows a long-tailed distribution, the number of samples per class differs—was this imbalance accounted for or corrected when computing class-wise FID? Lastly, since contrastive learning is often sensitive to batch size, it would be valuable to include an ablation study examining the impact of batch size on performance.

**Ethical Concerns:**

["NO or VERY MINOR ethics concerns only"]

**Final Justification:**

The concerns I raised were appropriately addressed through the rebuttal and additional experiment of the author. So I raising my score to 4.

**Limitations:**

yes

**Paper Formatting Concerns:**

There is no major formatting issues.

**Quality:**

3

**Strengths And Weaknesses:**

Strengths
- One of the key strengths of this paper is its simplicity and effectiveness: by appending only a lightweight linear projection head to the encoder bottleneck of the U-Net and applying a supervised contrastive loss, the authors successfully disentangle latent representations across classes. This disentanglement alleviates the representation entanglement issue commonly observed in diffusion models trained on long-tailed datasets, particularly benefiting tail-class generation. The impact of this regularization is well-supported both numerically (e.g., improved FID, IS, Recall, F1/8 in Table 1 and 3) and visually (e.g., Figure 1, 2, 10, and 13), where t-SNE and UMAP plots demonstrate clear class-wise separation in the latent space.

Weaknesses
- Although CORAL is primarily evaluated under long-tailed data settings, its core mechanism—contrastive latent regularization—seems broadly applicable beyond class imbalance. This raises the question of whether the problem formulation should have been positioned more generally as a method for representation disentanglement in diffusion models. Broadening the scope could improve both the conceptual clarity and the perceived applicability of the method.

- The paper lacks a detailed explanation of how class-wise latent disentanglement directly leads to improved sample quality, particularly in terms of FID. Introducing regularization in the latent space could, in theory, interfere with the likelihood-based training objective of DDPM and potentially steer the model away from the true data distribution. A more thorough analysis or targeted ablations would help clarify how the proposed regularization supports or enhances generative performance.

- Additionally, the paper adopts classifier-free guidance (CFG) during sampling but does not specify how the class label $y$ is determined at inference time. This omission is critical, as $y$ directly influences the conditional generation process and affects class-specific evaluations such as per-class FID and qualitative comparisons (e.g., Figures 3, 4, and 15). Without a clear description of how $y$ is chosen, the reproducibility and interpretability of the results may be compromised.

- Finally, while the method is effective, the idea of applying contrastive loss for latent regularization may appear somewhat marginal or incremental, given its conceptual similarity to existing contrastive learning techniques in other generative frameworks. Additional novelty or theoretical grounding could help distinguish it further from prior work.

---

> ### Author Rebuttal · Authors · 2025-07-31
>
> We sincerely thank the reviewer for their insightful feedback and thorough evaluation. Thank you for acknowledging our simplicity and effectiveness and recognizing that our approach successfully disentangles latent representations.  The recommended additions have helped address the identified limitations and provided deeper insights into our method’s behavior. These improvements positively enhance our contribution's impact. Below, we carefully address each concern, referring to weaknesses as W(i) and addressing the questions:
>
>
> - **W1** Thank you for this insightful observation about the broader applicability of our contrastive latent regularization approach. You are correct that the core mechanism could potentially benefit representation learning in diffusion models more generally. However, we specifically focus on long-tailed settings because these scenarios exhibit the most severe and well-documented problems with both fidelity and diversity of generated samples, particularly for tail classes.
> Contrastive latent regularization provides clear, measurable benefits in long-tail settings. While we agree that exploring the broader applicability to general representation disentanglement would be valuable future work, we hope that our demonstration of the effectiveness of contrastive latent regularization in this challenging scenario will inspire future research to explore its potential in more general diffusion model settings and broader representation learning contexts.
>
> - **W2**
> Thank you for this important question about how our latent space regularization enhances rather than interferes with generative performance. We address this concern through both theoretical grounding and comprehensive empirical validation.
> CORAL learns separation rather than imposed constraints on the generative process. While our contrastive regularization does affect the diffusion learning process, learned metric embeddings through projection heads enable the model to develop inherently separated class representations that support better generative performance.
> Recent theoretical analysis has demonstrated that contrastive losses with projection heads capture key task-related information more effectively than direct feature space constraints. The projection head enables better representation learning by decoupling the contrastive objective from the main diffusion features, allowing the model to learn disentangled representations in a auxiliary embedding space while the main bottleneck features continue to serve the diffusion objective. This complementary learning approach results in improved rather than competing objectives.
> Our comprehensive experiments across multiple datasets provide strong empirical evidence that latent regularization enhances rather than degrades generative performance. We consistently observe improvements in all generation metrics (FID, IS, Recall), suggesting that better-separated representations actually enhance the diffusion training objective.
>
> - **W3, Details on experimental setup** Thank you for this important clarification request regarding our experimental setup. We provide the following details and we will include all details in the final submission:
>   1. During sampling for generated data, we sample the class label $y$ from a uniform distribution across all classes, ensuring equal representation regardless of the training distribution imbalance. This approach provides a fair comparison across methods by evaluating each method's ability to generate high-quality samples for all classes, including underrepresented ones.
>   2. For overall metric calculations  we use the balanced version of the datasets for the real data to ensure fair evaluation. All metrics are measured with 50K generated images, except for per-class FID studies. For per-class FID analysis (Figure 3 and Figure 8), we generate 5K samples for each class and compare against 5K real samples from the balanced dataset for that specific class.
>   3. We tune the classifier-free guidance strength $\omega$ for both baselines and our method to ensure optimal performance for each approach.
>   4. Details on hyperparameter settings are provided in Table 2 in the appendix.
> This experimental design is used in state of the art and allows us to fairly evaluate each method's capability to generate high-quality samples across all classes, particularly focusing on the challenging tail classes.
>
> - **W4** While the use of contrastive learning and projection heads are established techniques in representation learning [1, 2], their application to minority class generation in diffusion models is novel because we are the first to identify latent representation entanglement at the U-Net bottleneck as a key mechanism underlying poor minority class generation. Previous work on improving long-tailed generation [3,4,5] has focused on ambient space solutions, missing this critical issue occurring within the diffusion architecture itself. Our key contributions are three-fold: (1) identifying representation entanglement at the U-Net bottleneck as the core failure mode, (2) recognizing that effective intervention requires \textit{learning} inherently separated representations rather than simply enforcing separation in the ambient space, and (3) demonstrating that the projection head is essential for creating a learned embedding space optimized for class discrimination through contrastive objectives.
> The necessity of projection heads and supervised contrastive learning has been well-established in other domains. It has been demonstrated that projection heads are crucial for contrastive learning [1, 6], showing that removing them significantly degrades representation quality and class separation. In [2], the authors provided theoretical analysis showing that projection heads help decouple the contrastive objective from the downstream task representation. Additionally, our choice of supervised contrastive learning is motivated by its demonstrated advantages [6]: the ability to leverage multiple positives per class within each batch, superior performance in learning discriminative representations, and better class boundary separation, which directly addresses our goal of reducing representation entanglement in long-tailed diffusion models.
> However, we are the first to provide empirical evidence of these principles specifically for generative diffusion models in long-tailed settings. As demonstrated in our ablation study (Figure 9, left panel), lower temperatures ($\tau_\mathrm{sc} \simeq 0$) achieve optimal performance by increasing sensitivity to hard positive and negative examples within each batch. We will add a ablation study examining the interaction between batch size and performance in our final submission.
> We verified that direct contrastive loss on bottleneck features as well as ambient-space approaches fail to achieve comparable improvements, and will add those ablation studies to the final submission. Our work establishes the empirical foundation for contrastive regularization in diffusion models for long-tailed image generation
>
>
> - **Sensitivity to batch size** This is an excellent question about batch size sensitivity in contrastive learning, and we appreciate the reviewer bringing up this important consideration.
> We would like to note that our choice of temperature $\tau_{SC}$ (Figure 9 on appendix, left panel) in the SupCon loss is motivated by the results in the appendix of  [6] which show that lower temperatures focus learning on hard positives/negatives, improving the discriminative power of learned representations even when working with the available examples in each batch.
> We acknowledge that a comprehensive ablation study examining the interaction between batch size and performance would strengthen our analysis and provide valuable insights into the robustness of our approach. We will include this experiment in the final submission.
>
> -**Additional Results:** We have expanded the scope of our experiments to include ImageNet-LT, please see a table with results in our response to the weakness stated by Reviewer aGJo.
>
> ---
>
> We thank you for the constructive feedback on our work. Working on the pointers has helped us improve the quality of our analysis. We hope we were able to clarify all your concerns, and look forward to resolving any remaining concerns during the discussion phase.
>
>
> [1] T. Chen, S. Kornblith, M. Norouzi, and G. Hinton. A simple framework for contrastive learning of visual representations, 2020.
>
> [2] Y. Xue, E. Gan, J. Ni, S. Joshi, and B. Mirzasoleiman. Investigating the benefits of projection head for representation learning, 2024.
>
> [3] Y. Qin, H. Zheng, J. Yao, M. Zhou, and Y. Zhang. Class-balancing diffusion models. In Proceedings of the IEEE/CVF Conference on Computer Vision and Pattern Recognition, 2023.
>
> [4] T. Zhang et al. Long-tailed diffusion models with oriented calibration, 2024.
>
> [5] D. Yan, L. Qi, V. T. Hu, M.-H. Yang, and M. Tang. Training class-imbalanced diffusion model via overlap optimization. CoRR, abs/2402.10821, 2024.
>
> [6] P. Khosla, P. Teterwak, C. Wang, A. Sarna, Y. Tian, P. Isola, A. Maschinot, C. Liu, and D. Krishnan. Supervised contrastive learning. In Advances in Neural Information Processing Systems, volume 33, pages 18661–18673, 2020.

---

> > ### Comment · Senior_Area_Chairs · 2025-08-04
> > **Please react to rebuttal**
> >
> > Dear Reviewer,
> >
> > The window to interact with authors is closing. Please take a look at their rebuttal ASAP.

---

> ### Comment · Reviewer_rXbT · 2025-08-04
>
> Thank you for addressing my concerns. I now fully understand the authors' claim regarding the benefit of modifying the U-Net bottleneck for long-tail datasets. The additional experiments provided in response to reviewer aGJo also helped clarify the effectiveness of the proposed method. Based on this, I'll increase my score to 4.

---

> > ### Author Response · Authors · 2025-08-05
> >
> > We greatly appreciate the reviewer's thoughtful reconsideration. We thank the reviewer for taking the time to review our responses and additional materials. The reviewer's insights have been quite helpful in ensuring that our work is clearly presented and well-supported.

---

### Official Review · Reviewer_AH8g · 2025-07-01

**Clarity:** 3
**Significance:** 2
**Originality:** 2
**Rating:** 4
**Confidence:** 4

**Summary:**

The paper proposes a method for improving the generations of tail classes. The method is motivated by the observation that in the bottleneck features of the diffusion model, the latent representations of classes of the head and tail classes overlap. The method involved training the diffusion model with a contrastive loss on the bottleneck features. Supportive results were shown on several long-tailed datasets.

**Questions:**

- In Fig 3, why is the performance of DDPM poor for all classes? I would expect that it would perform similarly to the baseline for the head class. Which is the head/tail class in this example?
- With regards to the last point above, did the authors observe a decrease in the diversity of the generations in the non tail classes with the contrastive loss?
- In what settings would such methods (training a custom diffusion model from scratch on long-tailed data), be preferable to controlling a pre-trained model like Stable Diffusion via e.g., prompt engineering, textual inversion?

**Ethical Concerns:**

["NO or VERY MINOR ethics concerns only"]

**Final Justification:**

I appreciate the authors rebuttal and the additional results. They have addressed my concerns. I will be increasing my score.

**Limitations:**

yes

**Quality:**

2

**Strengths And Weaknesses:**

**Strengths**

- The paper was well written and easy to follow.

**Weakness**

- There could be more thorough experiments to justify key design choices of the method. Several components of the method, e.g., where to apply the loss (e.g., bottleneck features vs. denoised images), or the selected loss function are used without clear motivation or empirical evidence. E.g., it is not clear why applying the contrastive loss at the bottleneck layer is preferred over applying it directly to the denoised output, similar to DiffROP [4]. The choice of contrastive loss over simpler alternatives such as reweighting schemes is not well-justified.
- The paper highlights the overlap between head and tail classes in the bottleneck features. However, DiffROP made the observation of an overlap between the tail and head classes although in the image space. It makes sense that in some intermediate layer their features also overlap.
- It seems like the contrastive loss could reduce the diversity of the generations within each class. This can also be seen from Fig. 1.

---

> ### Author Rebuttal · Authors · 2025-07-31
>
> We sincerely thank you for your thoughtful feedback and for finding our paper well written. We have addressed each concern comprehensively strengthen both our empirical validation and theoretical motivation. Below we detail our responses using W(i) and Q(i) for weakness and question, respectively:
>
> - **W1:** Our choice of bottleneck intervention is theoretically and empirically motivated over ambient approaches. Contrastive losses with projection heads capture task-related information more effectively [1,3], with [3] showing projection heads decouple contrastive objectives from downstream task representations. Our approach learns disentangled representations at the semantically rich bottleneck layer. Following the theoretical framework established in [1,2,3], the projection head facilitates learning of inherently separated class representations.
> Our empirical results provide strong evidence supporting this theoretical framework. Our extended experiments across multiple datasets consistently demonstrate that:
>   1. Contrastive loss in ambient space on negative pairs only (similar to DiffROP) can reduce intra-class diversity
>   2. Direct contrastive loss on bottleneck features without projection heads can affect the diffusion objective
>   3. Bottleneck contrastive learning with projection heads (CORAL) significantly outperforms ambient methods on generation metrics.
>     We will include these comprehensive ablation studies in the final submission to demonstrate the limitations of existing ambient-space approaches and showing that effective intervention requires learning inherently separated representations rather than simply enforcing separation in the ambient space.
>     Our architectural choices are grounded in established contrastive learning principles from [1], which demonstrated that simple linear projection heads with normalization are optimal for most contrastive learning tasks. Our experiments confirmed that this architecture transfers effectively to the diffusion setting, we will include our ablation studies on projection head architecture in the final submission.
>     Our choice of $\lambda(t)$ in Eq. (7) ensures that the contrastive loss weight increases as we approach the original data ($t \rightarrow 0$), where semantic structure is most recoverable and class discrimination most meaningful. The exponential decay provides smooth, differentiable weighting that doesn't introduce training instabilities. Recent work in diffusion models have implemented this [4,5], as different timesteps require different levels of structural guidance.
> Additionally, please see ImageNet-LT results in our response to Reviewer aGJo.
>
> - **W2:** Reweighting approaches (CBDM[6], TIW[4]) face limitations:
>     1. Estimation challenges - accurately estimating class ratios is difficult and error-prone [7], with TIW requiring additional discriminator training that introduces computational overhead.
>     2. Training instability - heavy reweighting of tail classes can cause gradient explosion, particularly problematic for diffusion models.
>     3. Mode collapse risk - over-emphasizing tail classes reduces sample diversity, as noted in CBDM.
>     Additionally, CBDM's reliance on head class predictions to enhance tail generation introduces bias and reduces robustness as noted in [8].
>     Our experimental results demonstrate that CORAL's representation-based approach consistently outperforms these reweighting methods across all datasets and metrics presented.
>
> - **W3:** We thank the reviewer for this observation. The critical distinction is not simply where overlap occurs, but where contrastive intervention is most effective for learning disentangled representations.
>
>     While overlap may be observable at multiple levels (ambient space, intermediate layers, bottleneck), the effectiveness of contrastive learning depends critically on where the intervention is applied. Our experimental results demonstrate that applying contrastive loss at the bottleneck is optimal compared to interventions at other locations. This aligns with established principles from representation learning showing that contrastive objectives work best when applied to semantically rich, compressed representations [1,2].
>
>     We will include comprehensive ablation studies in the final submission showing: (1) Direct comparison with ambient space contrastive approaches, showing superior performance of bottleneck intervention, (2) Ablations demonstrating that projection heads are essential for effective contrastive learning at the bottleneck level.
>
> - **W4,Q2:** We thank the reviewer for this important observation. Our approach enhances rather than reduces within-class diversity through our use of a projection head. Recent theoretical work by [3] demonstrates that projection heads prevent class collapse and preserve within-class diversity by enabling pre-projection representations to learn subclass-level features that are not represented post-projection, thus maintaining diversity within classes.
>     Figure 1 demonstrates inter-class separation rather than reduced intra-class diversity - the t-SNE visualization shows CORAL separates previously entangled classes while each cluster retains internal structure. Figure 4 provides visual evidence that CORAL generates more diverse tulips with varied characteristics compared to competing methods.
>     Quantitatively, CORAL consistently outperforms baselines on diversity metrics (Recall in Table 1), demonstrating that our approach enhances rather than reduces within-class diversity, as established theoretically in [3].
>
> - **Q1:** Thank you for this important question about Figure 3. You are absolutely correct to expect different performance patterns between head and tail classes in a more severely imbalanced setting.
>     In Figure 3, we show results for CIFAR10-LT with an imbalance factor $\rho$ = 0.01. The classes are ordered alphabetically in the long-tail distribution: airplane (head class), auto, bird, cat, deer, dog, frog, horse, ship, and truck (tail class).
>     The reason DDPM shows relatively uniform performance across all classes in Figure 3 is that the imbalance factor $\rho$ = 0.01, while significant, is not extreme enough to create the level of performance degradation typically observed in severely imbalanced scenarios. The choice of the long-tail parameter $\rho$ directly affects the magnitude of per-class FID differences. With less severe imbalance, we observe more comparable FID scores across classes.
>     Figure 8 shows the same CIFAR10-LT dataset but with a much more severe imbalance factor of $\rho$ = 0.001. Figure 8 demonstrates the expected long-tail generation behavior you mentioned, we observe the expected behavior for long-tail generation problems, where tail classes exhibit substantially higher FID scores compared to head classes. We will include additional visualizations in the final submission showing this enhanced diversity across multiple classes.
>
>     This comparison between Figures 3 and 8 illustrates how the severity of class imbalance (controlled by $\rho$) directly influences the magnitude of performance differences across the head-to-tail spectrum, and demonstrates that CORAL's improvements are consistent across different levels of imbalance severity.
>
> - **Q3:** Thank you for this excellent question about the practical applicability of our approach versus using pretrained models.
>     Prompt engineering and textual inversion primarily help with diversity by forcing the model to sample from minority classes more frequently, but they don't inherently improve the quality of those minority class generations. These methods essentially guide the sampling process toward underrepresented regions of the learned distribution. We acknowledge that for many practical applications, especially those involving natural images with good textual descriptions, leveraging pretrained models with prompt engineering or textual inversion can be more efficient and effective. CORAL could very well be combined with these prompt-based methods to produce even better quality images, which represents an interesting direction for future research.
>
> ---
>
> Once again, we thank you for the constructive feedback on our work. Working on the pointers has helped us improve the quality of our analysis. We hope we were able to clarify all your concerns, and look forward to resolving any remaining concerns during the discussion phase.
>
> [1] T. Chen, S. Kornblith, M. Norouzi, and G. Hinton. A simple framework for contrastive learning of visual representations, 2020.
> [2] P. Khosla, P. Teterwak, C. Wang, A. Sarna, Y. Tian, P. Isola, A. Maschinot, C. Liu, and D. Krishnan. Supervised contrastive learning. In Advances in Neural Information Processing Systems, volume 33, pages 18661–18673, 2020.
> [3] Y. Xue, E. Gan, J. Ni, S. Joshi, and B. Mirzasoleiman. Investigating the benefits of projection head for representation learning, 2024.
> [4] Y. Kim, B. Na, M. Park, J. Jang, D. Kim, W. Kang, and I.-C. Moon. Training unbiased diffusion models from biased dataset. In The Twelfth International Conference on Learning Representations, 2024.
> [5] R. Baptista, A. Dasgupta, N. B. Kovachki, A. Oberai, and A. M. Stuart. Memorization and regularization in generative diffusion models, 2025.
> [6] Y. Qin, H. Zheng, J. Yao, M. Zhou, and Y. Zhang. Class-balancing diffusion models. In Proceedings of the IEEE/CVF Conference on Computer Vision and Pattern Recognition, 2023.
> [7] B. Rhodes, K. Xu, and M. U. Gutmann. Telescoping density-ratio estimation. In H. Larochelle, M. Ranzato, R. Hadsell, M. Balcan, and H. Lin, editors, Advances in Neural Information Processing Systems, volume 33, pages 4905–4916. Curran Associates, Inc., 2020.
> [8] T. Zhang, H. Zheng, J. Yao, X. Wang, M. Zhou, Y. Zhang, and Y. Wang. Long-tailed diffusion models with oriented calibration. In The Twelfth International Conference on Learning Representations, 2024.

---

> ### Author Response · Authors · 2025-08-05
> **Follow-up on Rebuttal Submission**
>
> We thank the reviewer again for their thorough review and consideration. We wanted to follow up on the rebuttal we submitted in response to their review. We have addressed the concerns and questions raised and would appreciate any feedback on whether our responses are satisfactory. The reviewer's insights have been quite helpful in ensuring that our work is clearly presented and well-supported. If any aspects of our rebuttal require further clarification or if there are additional questions, we would be happy to provide more details.

---

> > ### Comment · Reviewer_AH8g · 2025-08-07
> >
> > ---
> >
> > Thank you for the detailed clarifications.
> >
> > Q1. The results in Fig 8 makes sense to me. Although for Fig 3, it is still not clear why the performance for DDPM is much worse than CORAL for the head class (airplane), assuming the experimental settings are the same, when CORAL is built on DDPM.
> >
> > Q2. I find it hard to evaluate the visuals in Fig 4, also it seems some of the generations in all 3 methods, do not look like tulips. Furthermore, unless I misunderstood, the recall metric does not explicitly measure diversity, but rather the overall coverage of the generated samples.
> >
> > Q3. That seems to indicate that CORAL could be beneficial for more specialised (e.g., medical) domains where pre-trained models are unlikely to be able to generate relevant images without significant engineering.
> >
> > I appreciate the clarifications, they will be useful additions to the paper. However, I am inclined to keep my score.

---

> > > ### Author Response · Authors · 2025-08-08
> > > **Response to Reviewer AH8g**
> > >
> > > We thank the reviewer for their continued engagement. We appreciate the opportunity to address
> > > their remaining concerns and hope to resolve any remaining questions.
> > >
> > > **Q1 (DDPM performance vs. CORAL):** We respectfully note that it is well-established in the
> > > literature that methods building upon DDPM can and often do outperform the base DDPM model.
> > > Both CBDM [1] and CORAL outperform DDPM across all classes, including head
> > > classes, demonstrating the effectiveness of these enhanced approaches. Even in balanced cases, both
> > > methods achieve lower FID scores than DDPM as can be seen in Table 3 in the appendix. For CORAL, this
> > > improvement stems from CORAL’s contrastive loss, which promotes disentanglement in the latent
> > > space even in balanced settings, as illustrated in Figure 10 in the appendix.
> > >
> > > Additionally, we want to note that per the latent space visualizations in Figures 1,10 and 13, under DDPM the entanglement between classes becomes progressively worse with increased imbalance and disproportionately affects tail classes, which is precisely the core issue our method addresses. This explains why CORAL's improvements are most significant for tail classes under higher imbalance.
> > >
> > > **Q2 (Recall metric and diversity assessment):** Thank you for this important clarification about
> > > diversity measurement. We’d like to address how our evaluation methodology captures diversity
> > > through multiple complementary approaches:
> > > 1.  Intermediate Feature-Based Diversity Assessment: Our diversity evaluation uses intermediate
> > > features from pre-trained networks. Specifically, the F8 metric uses InceptionV3 features to measure
> > > how well generated samples cover the real data manifold. The improved Recall metric employs k-
> > > nearest neighbor manifold estimation on VGG16 features, providing robust estimates of sample
> > > coverage.
> > > 2.  Visual Assessment: The qualitative comparisons in Figure 4 serve as complementary evidence
> > > for diversity. We acknowledge the reviewer’s concern about evaluating the visuals in Figure 4. It’s
> > > important to note that for CIFAR100-LT with an imbalance of $\rho =$0.01, there are only 5 images
> > > for the tail classes, including the tulip class. The generated tulips represent variations from the
> > > training data, and we will provide the training images for the tulip class in our final submission to
> > > demonstrate that images generated by CORAL are consistent with the available training data. The
> > > visual assessment in Figure 4 demonstrates that CORAL generates tulips with diverse characteristics
> > > (varied scales, structures, backgrounds) compared to those generated by CBDM [1] and T2H [2] that show
> > > repetitive small flowers with grass backgrounds and reduced class fidelity. This visual diversity
> > > directly supports our quantitative findings.
> > > 3.  Lower CFG Guidance: Importantly, CORAL requires lower CFG guidance strengths compared to baseline methods, which is significant because higher CFG guidance has been shown to
> > > reduce diversity [3]. This represents a fundamental advantage - CORAL
> > > achieves better diversity not despite lower guidance, but because our latent space regularization
> > > reduces the need for strong ambient-space guidance that would otherwise reduce diversity.
> > >
> > > The combination of feature-based metrics (Recall, F8), distributional measures (FID, IS), and
> > > visual assessment captures both semantic and perceptual aspects of diversity. We welcome suggestions for additional diversity evaluation metrics that could further strengthen our assessment,
> > > particularly those that provide complementary perspectives.
> > >
> > > **Q3 (Applicability to specialized domains):** We thank the reviewer for recognizing the potential
> > > applicability of CORAL to specialized domains such as medical imaging. Although we agree that
> > > this represents a valuable direction, we respectfully note that potential broader applicability is
> > > typically considered a strength rather than a limitation in methodological contributions.
> > >
> > > We hope these clarifications address the concerns of the reviewer. Our core contributions remain technically sound and empirically validated across multiple datasets and metrics: identifying latent representation entanglement as a failure mode in long-tailed diffusion models and demonstrating that contrastive regularization at the bottleneck effectively addresses this issue.
> > >
> > > ---
> > >
> > > 1. Y. Qin, H. Zheng, J. Yao, M. Zhou, and Y. Zhang. Class-balancing diffusion models. In Proceedings
> > > of the IEEE/CVF Conference on Computer Vision and Pattern Recognition, 2023.
> > >
> > > 2. T. Zhang, H. Zheng, J. Yao, X. Wang, M. Zhou, Y. Zhang, and Y. Wang. Long-tailed diffusion
> > > models with oriented calibration. In The Twelfth International Conference on Learning Repre-
> > > sentations, 2024.
> > >
> > > 3. J. Ho and T. Salimans. Classifier-free diffusion guidance. In NeurIPS 2021 Workshop on Deep
> > > Generative Models and Downstream Applications, 2021

---

### Official Review · Reviewer_gbPZ · 2025-07-02

**Clarity:** 3
**Significance:** 2
**Originality:** 2
**Rating:** 4
**Confidence:** 4

**Summary:**

This paper addresses the challenge of generating high-quality, diverse images for underrepresented classes in long-tailed datasets using diffusion models. It identifies that poor performance on tail classes is due to representation entanglement—overlap in the latent space between head and tail classes within the U-Net bottleneck. To mitigate this, the authors propose CORAL (Contrastive Regularization for Aligning Latents), which adds a projection head to the bottleneck and applies a supervised contrastive loss to promote class separation. CORAL significantly improves tail-class generation across multiple datasets, outperforming prior methods both quantitatively and qualitatively.

**Questions:**

1. Is the contribution of CORAL substantial enough, given that it primarily repurposes existing contrastive learning techniques without introducing novel mechanisms?
2. Why does contrastive regularization in the latent space outperform ambient-space approaches, and can this be theoretically justified beyond empirical intuition?
3. How sensitive is CORAL to architectural and hyperparameter choices, and can more comprehensive ablations improve confidence in its robustness?
4. Do the generated tail-class images accurately reflect their intended semantics, and how well does CORAL preserve class fidelity under conditional generation?

**Ethical Concerns:**

["NO or VERY MINOR ethics concerns only"]

**Final Justification:**

The authors’ responses have addressed some of my concerns; however, I remain concerned about the lack of experiments on large-scale models and datasets. Therefore, I have decided to maintain my original score.

**Limitations:**

yes

**Paper Formatting Concerns:**

not found

**Quality:**

3

**Strengths And Weaknesses:**

**Strengths**

The paper makes a strong empirical observation that latent feature overlap in diffusion models is a core failure mode for tail-class generation under long-tailed distributions. This is a meaningful contribution that deepens our understanding of diffusion behavior. CORAL introduces a straightforward architectural addition (a projection head with contrastive loss) to existing diffusion frameworks, showing consistent improvements across multiple datasets and metrics, without requiring external models or complex rebalancing schemes.

Weaknesses

* Limited methodological novelty: The main contribution lies in *where* contrastive learning is applied (i.e., the bottleneck of U-Net), rather than in the contrastive mechanism itself. The approach builds on well-established tools (e.g., SupCon loss, projection head), making the core technique relatively incremental.

* Shallow theoretical grounding: The paper lacks theoretical justification or formal analysis of why contrastive regularization in latent space works better than in the ambient space. The intuition is sound but remains at an empirical level.

* Underdeveloped ablation and sensitivity analysis**: While some ablations are mentioned, there is limited exploration of how sensitive CORAL is to design choices like the projection head architecture, temperature hyperparameters, or the exact form of φ(t). This weakens the assessment of robustness.

* No analysis of class-conditional generation quality: The evaluation is mostly focused on overall metrics and per-class FID. It does not deeply assess whether generated tail-class images are semantically correct or align well with class labels, particularly under CFG.

---

> ### Author Rebuttal · Authors · 2025-07-31
>
> We thank the reviewer for their detailed evaluation. Thank you for recognizing our strong empirical observation and meaningful contribution that deepens understanding of diffusion behavior. Your feedback has provided an excellent opportunity to explain our main contributions more clearly. We have justified our design choices with additional experiments and theoretical motivation. The added details demonstrate how our work provides novel insights and demonstrates improved performance. Below we detail our responses using W(i) and Q(i) for weakness and questions, respectively:
>
> - **W1,Q1:**
>   While the use of contrastive learning and projection heads are established techniques in representation learning [1, 2], their application to minority class generation in diffusion models is novel because we are the first to identify latent representation entanglement at the U-Net bottleneck as a key mechanism underlying poor minority class generation. Previous work on improving long-tailed generation has focused on ambient space solutions, missing this critical failure mode. Our key contributions are three-fold:
>   (1) identifying representation entanglement at the U-Net bottleneck as the core failure mode,
>   (2) recognizing that effective intervention requires learning inherently separated representations rather than simply enforcing separation in the ambient space, and
>   (3) demonstrating that the projection head is essential for creating a learned embedding space optimized for class discrimination through contrastive objectives.
> The advantages of projection heads and supervised contrastive learning has been well-established in other domains. It has been demonstrated that projection heads are crucial for contrastive learning [1, 3], showing that removing them significantly degrades representation quality and class separation. In [2], the authors provided theoretical analysis showing that projection heads help decouple the contrastive objective from the downstream task representation. Additionally, our choice of supervised contrastive learning is motivated by its demonstrated advantages [3]: the ability to leverage multiple positives per class within each batch, superior performance in learning discriminative representations, and better class boundary separation, which directly addresses our goal of reducing representation entanglement in long-tailed diffusion models.
> However, we are the first to provide empirical evidence of these principles specifically for generative diffusion models in long-tailed settings. As demonstrated in our ablation study (Figure 9, left panel), lower temperatures ($\tau_\mathrm{sc} \simeq 0$) achieve optimal performance by increasing sensitivity to hard positive and negative examples within each batch. We will add an ablation study examining the interaction between batch size and performance in our final submission.
> We verified that direct contrastive loss on bottleneck features (without a projection head) as well as ambient-space approaches fail to achieve comparable improvements, and will add those ablation studies to the final submission. Our work establishes the empirical foundation for contrastive regularization in diffusion models for long-tailed image generation.
>
> - **W2,Q2:**
>   Our approach is grounded in both established representation learning principles from other encoder settings and our experimental results that demonstrate latent contrastive loss with projection heads significantly outperforms ambient contrastive approaches for diffusion models. Recent theoretical analysis has shown that contrastive losses with projection heads capture key task-related information more effectively than direct feature space constraints in classification and self-supervised learning [1, 2].
> We have conducted extensive experiments that provide empirical validation across multiple long-tailed datasets confirming that these principles extend to generative diffusion models, and showing that latent space intervention with projection heads consistently improves all generation metrics (FID, IS, Recall) compared to ambient space methods. We will include these results in the final submission.
> The theoretical foundation for why latent space intervention outperforms ambient space approaches lies in the fundamental difference between learned versus imposed separation. Our approach addresses the root cause by learning disentangled representations at the semantically rich bottleneck layer where class overlap occurs. This is theoretically sound because:
>   (1) the bottleneck contains compressed semantic information before decoder expansion, making it the optimal intervention point, and
>   (2) learned metric embeddings through projection heads enable the model to develop inherently separated class representations rather than merely correcting entangled outputs.
> Our empirical results provide strong evidence supporting this theoretical framework. Our extended experiments across multiple datasets consistently demonstrated that:
>   1. Contrastive loss in ambient space does not improve or actually reduces diversity (lower recall),
>   2. Direct contrastive loss on bottleneck features without projection heads is suboptimal, and
>   3. Bottleneck contrastive learning with projection heads significantly improves all generation metrics.
>   We will include these comprehensive ablation studies in the final submission to demonstrate the limitations of existing ambient-space approaches and validate our latent space intervention strategy.
> We have also performed extensive ablation studies on the projection head architecture that we will include in the final submission. Our architectural choices are grounded in established contrastive learning principles from [1], which demonstrate that simple linear projection heads with normalization are optimal for most contrastive learning tasks. Our experiments confirm that this architecture transfers effectively to the diffusion setting.
>
> - **W3,Q3:**
>   We have included comprehensive ablation studies capturing the dependence of FID on key hyperparameters in Fig. 9 in the appendix. We present three plots showcasing the dependence of FID on $\tau_r$ from Eqn. (7) (the contrastive loss decay rate), $\tau_\mathrm{sc}$ from Eqn. (5) (the SupCon temperature), and the CFG weight, $\omega$. The FID is plotted for values around the found minimum, demonstrating CORAL's robustness across reasonable parameter ranges.
>   The functional form $\lambda(t) = w \cdot \exp \left( \frac{1 - t/T}{\tau_r} \right)$ ensures that:
>   (1) the contrastive loss weight increases as we approach the original data ($t \rightarrow 0$), where semantic structure is most recoverable and class discrimination most meaningful,
>   (2) the exponential decay provides smooth, differentiable weighting that doesn't introduce training instabilities.
>   Recent work on diffusion models has explored adapting regularization strength based on noise levels [4, 5], recognizing that different timesteps require different levels of structural guidance.
> We have also performed extensive ablation studies on the projection head architecture that we will include in the final submission. Our architectural choices are grounded in established contrastive learning principles from [1], which demonstrate that simple linear projection heads with normalization are optimal for most contrastive learning tasks. Our experiments confirm that this architecture transfers effectively to the diffusion setting.
>
> - **W4,Q4:**
>   Our results demonstrate strong class fidelity preservation through both quantitative and qualitative evidence. Figures 3 and 8 show per-class FID improvements across all classes in CIFAR10-LT ($\rho = 0.01$ and $\rho = 0.001$), with CORAL consistently outperforming baselines. Figure 4 provides compelling qualitative evidence of semantic preservation for the tulip class in CIFAR100-LT. We observe that CBDM and H2T suffer from feature borrowing — producing tulips with repetitive, grass-heavy backgrounds borrowed from dominant animal classes — while CORAL generates tulips with diverse, semantically appropriate backgrounds that reflect the true tulip class distribution.
> We will include additional qualitative comparisons in the final submission showing semantic preservation across multiple classes, providing visual evidence of improved class fidelity that complements our quantitative results.
>
> - **Additional Results:**
> We have expanded the scope of our experiments to include ImageNet-LT, please see a table with results in our response to the weakness stated by *Reviewer aGJo*.
> ---
>
> Once again, we thank you for the constructive feedback on our work. Working on the pointers has helped us improve the quality of our analysis. We hope we were able to clarify all your concerns, and look forward to resolving any remaining concerns during the discussion phase.
>
>
>
> [1] T. Chen, S. Kornblith, M. Norouzi, and G. Hinton. A simple framework for contrastive learning of visual representations, 2020.
>
> [2] Y. Xue, E. Gan, J. Ni, S. Joshi, and B. Mirzasoleiman. Investigating the benefits of projection head for representation learning, 2024.
>
> [3] P. Khosla, P. Teterwak, C. Wang, A. Sarna, Y. Tian, P. Isola, A. Maschinot, C. Liu, and D. Krishnan. Supervised contrastive learning. In Advances in Neural Information Processing Systems, volume 33, pages 18661–18673, 2020.
>
> [4] Y. Kim, B. Na, M. Park, J. Jang, D. Kim, W. Kang, and I.-C. Moon. Training unbiased diffusion models from biased dataset. In The Twelfth International Conference on Learning Representations, 2024.
>
> [5] R. Baptista, A. Dasgupta, N. B. Kovachki, A. Oberai, and A. M. Stuart. Memorization and regularization in generative diffusion models, 2025.

---

> ### Author Response · Authors · 2025-08-05
> **Follow-up on Rebuttal Submission**
>
> We thank the reviewer again for their thorough review and consideration. We wanted to follow up on the rebuttal we submitted in response to their review. We have addressed the concerns and questions raised and would appreciate any feedback on whether our responses are satisfactory. The reviewer's insights have been quite helpful in ensuring that our work is clearly presented and well-supported. If any aspects of our rebuttal require further clarification or if there are additional questions, we would be happy to provide more details.

---

> ### Comment · Reviewer_gbPZ · 2025-08-06
> **Response to authors**
>
> Thank you for your reply! I think my concerns about the theoretical part have been largely addressed, but the experimental part is still not convincing.
>
> **How do you fairly compute the per-class FID?** As we all know, when the number of examples is small, the estimation of distribution becomes inaccurate, and therefore, under these circumstances, FID will become larger and the results will not be reliable. From what I can see, your per-class FID is meaningless without more details.
>
> Another issue is that you have shown results for ImageNet-LT in your response to aGJo. The experimental details are missing, including which model was used and training details. I can responsibly tell you that I have experimented with models ranging from DiT-S (with or without CFG) to DiT-XL/2 (with or without CFG) on ImageNet-LT, and I roughly know the performance of models with similar parameters. **A very serious problem** is that, your result is totally not aligned with our expermental results. For example, for DiT-S, the results are as follows and can be easily reproduced using the DiT code. The evaluation toolkit is aligned with that of DiT. It's clear that **a model with such a low FID (in your experiment) cannot have such a low IS**.
>  I believe there is some training or evluation details you don't point out, but I think it's a serious problem.
>
> | Model | FID | sFID | IS | Precision | Recall |
> | --- | --- | --- | --- | --- | --- |
> | DiT-s (cfg) | 39.94 | 8.44 | 39.99 | 0.48 | 0.55 |
> | DiT-s (no cfg) | 63.93 | 11.99 | 22.04 | 0.37 | 0.56 |
>
> It would be better if you could include more details about **what kind of model you used, how many epochs you trained, and what dataset you used for evaluation** to make your results convincing.

---

> > ### Author Response · Authors · 2025-08-06
> > **Response to Reviewer gbPZ**
> >
> > We thank the reviewer for their detailed response and highlighting the importance of comprehensive experimental details. These details will be provided in our final submission, we address each of the reviewer's concerns below.
> >
> > **Per-class FID Computation:** For the per-class FID analysis on CIFAR10-LT (Figures 3 and 8), we generate 5K samples per class and compare against 5K real samples from the balanced dataset for each class. For CIFAR10, the balanced training data contains exactly 5K samples per class, thus the choice of 5K samples for per-class FID computation following established practices [1,2].
> >
> > **Experimental Details:** We use the DDPM model from Ho et al. [3], consistent with the methods we compare against. Our implementation follows that of CBDM [1], which is based on the publicly available implementation of DDPM, all details will be provided in the final submission. We use ImageNet-LT [4] at 64×64 resolution with batch size 128, training for 300K iterations. For evaluation, we generate 50K samples uniformly across all classes and compare against the balanced validation set containing 20K images as the real samples. We re-implemented all baseline methods under these identical conditions to ensure fair comparison and eliminate confounding variables, which explains why our results may differ from those reported in papers using different setups. The ImageNet-LT results under our experimental framework are consistent with those of the comparison methods [2] and with prior work on class-imbalanced generation in GANs [5].
> >
> > **Direct comparison with DiT results:** As demonstrated in prior work, including [6], FID and IS capture different aspects of generation quality and are highly dependent on the model and data. Although we would expect that a lower FID would correspond to higher IS under identical experimental conditions, a lower FID does not necessarily correspond to higher IS scores under different setups. Comparison with the reviewer's stated DiT results falls outside our current scope and is not an adequate benchmark without verified experimental alignment.
> >
> > **We agree that detailed explanation of datasets and evaluation methodology is essential for meaningful results.** We will include all experimental details in our final submission and welcome any specific suggestions for improving our evaluation methodology.
> >
> > ----
> > Once again, we thank the reviewer for their constructive feedback which helps us improve the quality of our analysis.
> >
> > 1. Y. Qin, H. Zheng, J. Yao, M. Zhou, and Y. Zhang. Class-balancing diffusion models. In Proceedings of the IEEE/CVF Conference on Computer Vision and Pattern Recognition, 2023.
> >
> > 2. T. Zhang, H. Zheng, J. Yao, X. Wang, M. Zhou, Y. Zhang, and Y. Wang. Long-tailed diffusion models with oriented calibration. In The Twelfth International Conference on Learning Representations, 2024.
> >
> > 3. J. Ho, A. Jain, and P. Abbeel. Denoising diffusion probabilistic models. In Proceedings of the 34th International Conference on Neural Information Processing Systems, 2020.
> >
> > 4. Z. Liu, Z. Miao, X. Zhan, J. Wang, B. Gong, and S. X. Yu. Large-scale long-tailed recognition in an open world. 2019 IEEE/CVF Conference on Computer Vision and Pattern Recognition (CVPR), pages 2532–2541, 2019
> >
> > 5. H. Rangwani, L. Bansal, K. Sharma, T. Karmali, V. Jampani, and R. V. Babu. NoisyTwins: Class-Consistent and Diverse Image Generation Through StyleGANs . In 2023 IEEE/CVF Conference on Computer Vision and Pattern Recognition (CVPR), pages 5987–5996, Los Alamitos, CA, USA, June 2023. IEEE Computer Society.
> >
> > 6. A. Borji. Pros and cons of gan evaluation measures. Computer Vision and Image Understanding, 179:41–65, 2019. ISSN 1077-31

---

> > > ### Comment · Reviewer_gbPZ · 2025-08-09
> > > **Response to authors**
> > >
> > > Thanks for your reply — the additional information you provided is detailed and convincing. I have decided to maintain my positive score for this paper.

---

### Note · Authors · 2025-08-15

We thank all reviewers for their constructive feedback and engagement throughout the review process. Our work makes three key contributions to long-tailed diffusion models:
1. Novel identification of latent representation entanglement at the U-Net bottleneck as a fundamental failure mode for tail-class generation.
2. CORAL framework that directly addresses this issue through contrastive regularization of bottleneck features, achieving consistent improvements across all evaluation metrics and datasets. The t-SNE visualizations show how CORAL achieves class separation.
3. Empirical validation demonstrating that contrastive latent regularization outperforms ambient space approaches, with detailed ablations confirming our architectural and hyperparameter choices.

We addressed all reviewer concerns with additional experiments in ImageNet-LT, detailed experimental methodology, and extensive ablation studies that will be included in the final submission. Our results demonstrate that CORAL provides an effective framework for addressing long-tailed generation.

---

### Decision · Program_Chairs · 2025-09-17

**Decision:**

Accept (poster)

**Comment:**

The authors propose CORAL (Contrastive Regularization for Aligning Latents), which introduces a projection head at the bottleneck and applies a supervised contrastive loss to separate head and tail representations to address the challenges that tail classes suffer from poor generative quality from diffusion models.

The paper identifies latent feature overlap in diffusion bottlenecks as a key cause of tail-class underperformance, which is novel and is less explored before, thus offering a valuable contribution to understanding diffusion behavior. As a solution, the authors propose a design with only a projection head and supervised contrastive loss. The authors dive deep in the empirical perspectives and provide sufficient evidence in different settings: comparing with previous works, CORAL show significant and consistent performance gain across class-long-tailed datasets, including CIFAR-10, CIFAR-100, ImageNet. The reviewers have spotted concerns regarding limited ablations and lack of theoretical contributions to fully reveal the properties.  In the rebuttal, the authors have provided comprehensive experiments to address the concerns. Although in the end the loss is designed from heuristic perspective, given the authors have provided sufficient empirical evidence, all reviewers agree the method is technical sound and give a positive rating. Currently the paper meets the bar of publication, and the authors are encouraged to clarify the exploration focus on the empirical perspectives by carefully revising the paper to incorporate the reviewers' comments, and continue the exploration on the theoretical side.